# Epigenome-wide DNA methylation association study of CHIP provides insight into perturbed gene regulation

Sara Kirmani[1,2,22], Tianxiao Huan[1,2,22] ✉, Joseph C. Van Amburg[3], Roby Joehanes[1,2], Md Mesbah Uddin[4,5], Ngoc Quynh H. Nguyen[6], Bing Yu[6], Jennifer A. Brody[7], Myriam Fornage[8,9], Jan Bressler[6,9], Nona Sotoodehnia[7], David A. Ong[3], Fabio Puddu[10], James S. Floyd[7,11], Christie M. Ballantyne[12], Bruce M. Psaty[7,11,13], Laura M. Raffield[14], Pradeep Natarajan[4,15,16], Karen N. Conneely[17], Joshua S. Weinstock[17], April P. Carson[18], Leslie A. Lange[19], Kendra Ferrier[19], Nancy L. Heard-Costa[1,20], Joanne Murabito[1,21], Alexander G. Bick[3,23] ✉ & Daniel Levy[1,2,23] ✉

With age, hematopoietic stem cells can acquire somatic mutations in leukemogenic genes that confer a proliferative advantage in a phenomenon termed CHIP. How these mutations result in increased risk for numerous age-related diseases remains poorly understood. We conduct a multiracial meta-analysis of EWAS of CHIP in the Framingham Heart Study, Jackson Heart Study, Cardiovascular Health Study, and Atherosclerosis Risk in Communities cohorts ($N = 8196$) to elucidate the molecular mechanisms underlying CHIP and illuminate how these changes influence cardiovascular disease risk. We functionally validate the EWAS findings using human hematopoietic stem cell models of CHIP. We then use expression quantitative trait methylation analysis to identify transcriptomic changes associated with CHIP-associated CpGs. Causal inference analyses reveal 261 CHIP-associated CpGs associated with cardiovascular traits and all-cause mortality (FDR adjusted $p$-value < 0.05). Taken together, our study reports the epigenetic changes impacted by CHIP and their associations with age-related disease outcomes.

A hallmark of aging is the accumulation of somatic mutations in dividing cells. The vast majority of these mutations do not affect cell fitness. In rare circumstances, however, a mutation can arise in a progenitor cell that confers a selective fitness advantage, culminating in its expansion relative to other cells. In the hematopoietic system, this process is termed clonal hematopoiesis (CH). Individuals with CH are at increased risk for the development of hematologic malignancies[1]. A subset of CH is driven by pathogenic mutations in myeloid malignancy-associated genes, which is termed CH of indeterminate potential (CHIP) and has been shown to be associated with hematologic cancers, cardiovascular disease (CVD), chronic obstructive pulmonary disease, and mortality, among other conditions[2–4].

The prevalence of CHIP increases with advancing age[2,5–7]. In a whole genome sequencing (WGS) study from the NHLBI Trans-Omics for Precision Medicine (TOPMed) program that included ~100,000 individuals across 51 separate studies, large CHIP clones were found to be uncommon (<1%) in individuals younger than 40 years of age and increased to 12% in those aged 70–89 and 20% in those aged 90 years and older[5]. This age-dependent pattern was consistent across CHIP driver genes[5] and has been observed in other studies[2,6,7].

DNA methylation (DNAm), the addition of a methyl group to a cytosine followed by a guanosine (CpG) in DNA, is an epigenetic modification that reflects age and environmental exposures. The gene products of the three most frequently mutated CHIP driver genes, *DNMT3A*, *TET2*, and *ASXL1*, are epigenetic regulators[5]. DNMT3A (DNA-methyltransferase 3A) is a methyltransferase that catalyzes the transfer of methyl groups to CpG sites and catalyzes de novo DNA methylation[8]. Conversely, TET2 (ten-eleven translocation-2) is a DNA demethylase that catalyzes the conversion of 5-methylcytosine to 5-hydroxymethylcytosine, one of the steps leading to eventual deme-thylation of CpG sites[9]. ASXL1 (ASXL transcriptional regulator 1) is involved in histone modification[10]. Its function in CHIP remains rela-tively unknown[11].

CHIP has been shown to be associated with global DNAm changes, particularly for the *DNMT3A* and *TET2* CHIP driver gene mutations[12]. A previous epigenome-wide association study (EWAS) of CHIP in 582 Cardiovascular Health Study (CHS) participants, with replication in 2655 Atherosclerosis Risk in Communities (ARIC) participants, revealed several thousand CpG sites associated with CHIP and its two major CHIP driver genes, *DNMT3A* and *TET2*[12]. *DNMT3A* and *TET2* CHIP were also found to have directionally opposing DNAm signatures: *DNMT3A* CHIP mutations were associated with hypomethylation of CpGs, whereas *TET2* CHIP was associated with hypermethylation of CpGs, consistent with the canonical regulatory functions of DNMT3A and TET2 elucidated in murine and human model systems[13–15].

Despite the wealth of information from the previous EWAS of CHIP[12], several limitations and knowledge gaps remain. These include the need to use larger sample sizes to enable analyses of less prevalent CHIP driver gene mutations such as *ASXL1*, explore downstream functions and pathways influenced by mRNA expression for any CHIP and CHIP subtypes, and identify underlying molecular mechanisms linking CHIP to CVD.

To address these knowledge gaps, we conduct a multiracial meta-analysis of separate EWAS of CHIP in four independent cohort studies (*N* = 8196; 462 with any CHIP, 261 *DNMT3A*, 84 *TET2*, and 21 with *ASXL1* CHIP) along with analysis of the associations of CHIP-related CpGs with downstream gene expression. We expand upon the previous EWAS of CHIP study[12] by adding two cohorts−the Framingham Heart Study (FHS) and the African-American Jackson Heart Study (JHS) – in addition to the ARIC and CHS cohorts. The EWAS findings are functionally validated using human hematopoietic stem cell (HSC) models of CHIP.

Expression quantitative trait methylation (eQTM) analysis identifies gene expression changes associated with CHIP-associated CpGs. Cau-sal inference analysis using two-sample Mendelian randomization (MR) is performed to gain insight into the molecular mechanisms linking CHIP to CVD. A flowchart of the study design is shown in Fig. 1.

## Results

### Clinical characteristics of study participants

The baseline characteristics of FHS, JHS, CHS, and ARIC participants included in this investigation are presented in Table 1. The mean age at the time of blood draw for whole-genome sequencing (WGS) was 57, 56, and 58 for FHS, JHS, and ARIC, respectively. Participants from CHS were considerably older, with a mean age of 74 years. All four cohorts had more women than men (54–63% women). Overall, CHIP mutations with a variant allele frequency (VAF) ≥ 2% were present in 5% (166/3295) of participants in FHS, 4% (68/1664) in JHS, 5% (142/2655) in ARIC, and 15% (86/582) in CHS. Consistent with previous reports[5], the three most frequently mutated CHIP driver genes across all cohorts were *DNMT3A*, *TET2*, and *ASXL1*. Eighty percent of individuals with CHIP demonstrated expanded CHIP clones with VAF > 10%.

### Epigenome-wide association analysis

Race was classified based on self-report. In the race-stratified analysis, we identified 2843 CpGs associated with any CHIP, 758 with *DNMT3A*, 4735 with *TET2* CHIP in White participants and 5498 with any CHIP, 5065 with *DNMT3A*, and 290 with *TET2* CHIP in Black participants at Bonferroni-corrected $P < 1 \times 10^{-7}$ (Supplementary Data 1–6). 1290, 675, and 254 CHIP-associated CpG sites were shared between White and Black participants at the Bonferroni-corrected threshold, with con-cordant directions of effect for any CHIP, *DNMT3A*, and *TET2* CHIP, respectively.

In a multiracial, meta-EWAS of CHIP, 9615 CpGs were associated with any CHIP, and 5990, 5633, and 6078 CpGs were associated with *DNMT3A* CHIP, *TET2* CHIP, and *ASXL1* CHIP, respectively (at Bonferroni-corrected $P < 1 \times 10^{-7}$). The top ten CpGs for any CHIP and for each of the three CHIP driver genes are shown in Table 2. A full list of CpG signatures and their directions of effect are reported in Sup-plementary Data 7-10. There was minimal to moderate overlap of CpGs associated with *DNMT3A*, *TET2*, and *ASXL1*; 429, 904, and 1088 CpG sites were shared between *DNMT3A* and *TET2*, *DNMT3A* and *ASXL1*, and *TET2* and *ASXL1*, respectively.

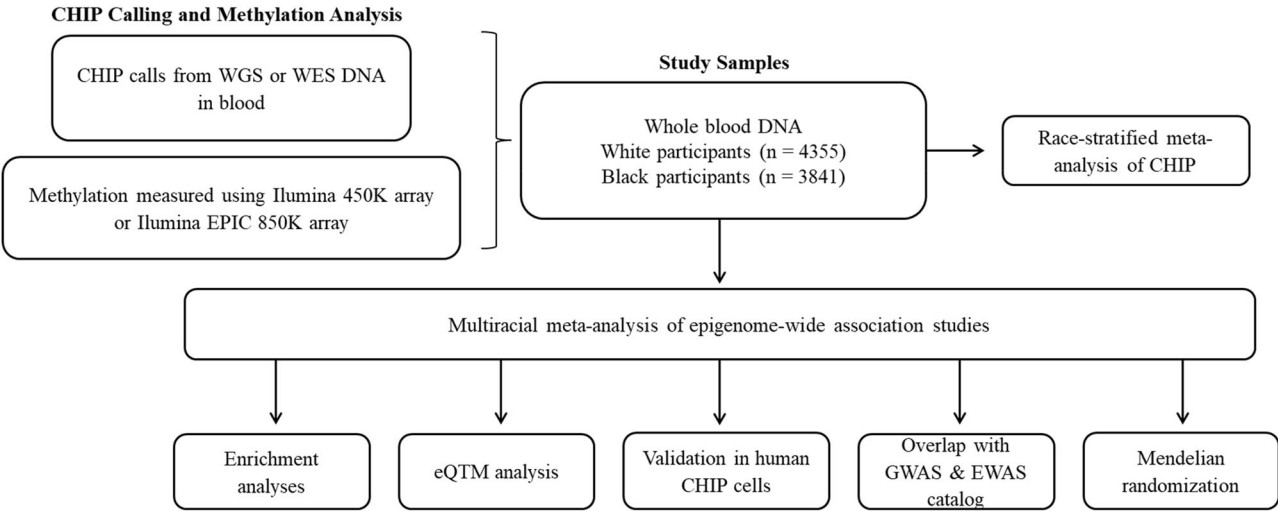

**Fig. 1 | Overview of Study Design.** This flowchart outlines the sequential steps of the study, from data collection to downstream analyses. *CHIP* Clonal Hematopoi-esis of Indeterminate Potential, *WGS* Whole Genome Sequencing, *WES* Whole Exome Sequencing, *TOPMed* Trans-Omics for Precision Medicine program, *eQTM* Expression Quantitative Trait Methylation.

**Table 1 | Baseline Characteristics of Cohorts**

| Study | N | CHIP cases, N | CHIP cases at VAF > 10% | *DNMT3A* CHIP | *TET2* CHIP | *ASXL1* CHIP | White partici-pants, N | Black partici-pants, N | Age, mean (range) | Sex, Female (%) | Smoking, N |
|---|---|---|---|---|---|---|---|---|---|---|---|
| FHS | 3295 | 166 | 145 (87%) | 77 (46%) | 38 (23%) | 21 (13%) | 3295 | 0 | 57 (24–92) | 54 | 343 |
| JHS | 1664 | 68 | 63 (93%) | 44 (65%) | 14 (21%) | N/A* | 0 | 1664 | 56 (22–93) | 63 | 236 |
| CHS | 582 | 86 | 76 (88%) | 35 (41%) | 18 (21%) | N/A* | 302 | 280 | 74 (64–91) | 61 | 320 |
| ARIC | 2655 | 142 | 86 (61%) | 105 (74%) | 14 (10%) | N/A* | 758 | 1897 | 58 (47–72) | 61 | 1486 |

*Less than 5 *ASXL1* CHIP cases for indicated cohorts.

*FHS* Framingham Heart Study, *JHS* Jackson Heart Study, *CHS* Cardiovascular Health Study, *ARIC* Atherosclerosis Risk in Communities, *CHIP* Clonal Hematopoiesis of Indeterminate Potential, *VAF* Variant Allele Fraction.

We identified 5987 CpGs (~100%) associated with *DNMT3A* CHIP and 4607 CpGs (~76%) associated with *ASXL1* CHIP that showed decreased methylation ($\beta < 0$) (Fig. 2b, d). In contrast, 5079 CpGs (~90%) associated with *TET2* CHIP showed increased methylation ($\beta > 0$) (Fig. 2c). Out of the 554 *TET2*-associated CpGs that showed decreased methylation, 171 (~31%) CpGs were found to overlap with *DNMT3A* CpG sites. The vast majority of CpGs associated with CHIP were remote (>1 Mb) from the driver gene including 5969/5990 (99.6%) for *DNMT3A*, 5632/5633 (~100%) for *TET2*, and 6070/6078 (99.9%) for *ASXL1*.

Although age was included as a covariate, there remains a possibility that common CpG sites across all three CHIP driver genes may be related to age rather than CHIP mutation. To assess this, we compared the 19 CpGs that are common among all three CHIP driver gene mutations from our meta-EWAS of CHIP with the CpGs from a recent EWAS of chronological age in the Generation Scotland cohort[16] (N = 18,413). No CpGs overlapped, suggesting that common CpGs across all CHIP driver genes are related to CHIP mutation rather than age.

A sensitivity analysis was performed by excluding CHIP cases with VAF < 10%. The results are similar to the multiracial meta-EWAS of any CHIP and are provided in Supplementary Fig. 4 and Supplementary Data 11-13. Approximately 78% of CpGs (7460/9615) in the meta-EWAS of any CHIP were re-identified in the sensitivity analysis, while 312 CpGs were newly identified.

**Human hematopoietic stem cell models of CHIP validate EWAS**
We sought to experimentally validate our multiracial meta-EWAS methylation findings with an in vitro model of CHIP. CHIP was modeled by introducing loss-of-function mutations in *DNMT3A*, *TET2*, and *ASXL1* in mobilized peripheral blood CD34+ hematopoietic cells, using CRISPR-Cas9[17]. After seven days in culture, these cells were flow sorted to isolate a purified population of CD34⁺CD38⁻Lin⁻ cells. Following fluorescence-activated cell sorting, genomic DNA (gDNA) was extracted, and methylation was assayed using biomodal duet evoC (see Methods)[18].

The analysis focused on the subset of CpG sites that were significantly associated with CHIP in the EWAS data and nominally differentially methylated ($P < 0.05$) in the in vitro model of CHIP. When comparing CpG site subsets with their respective engineered cells, *DNMT3A*-associated CpG sites showed significant enrichment in *DNMT3A*-engineered cells ($P < 2.88 \times 10^{-239}$) (Fig. 3A), while *TET2*-associated CpG sites were significantly enriched in *TET2*-engineered cells ($P < 8.39 \times 10^{-56}$) (Fig. 3B), and *ASXL1*-engineered cells ($P < 1.65 \times 10^{-14}$) (Supplementary Fig. 5). *ASXL1*-associated CpG sites showed no significant hits in *ASXL1*-engineered cells (Fig. 3C), but a slight trend in *TET2*-engineered cells (Supplementary Fig. 5). The any CHIP-associated CpG sites were significantly enriched in *DNMT3A*-engineered primary cells only ($P < 9.29 \times 10^{-121}$), unlike *TET2* and *ASXL1* (Supplementary Fig. 5).

**CpG association with gene expression and pathway analyses**
To investigate the functional consequences of CHIP-associated CpGs, we performed gene ontology (GO) and pathway enrichment analysis for genes harboring CHIP-associated CpGs. For any CHIP, *DNMT3A* CHIP, and *ASXL1* CHIP, the enriched GO terms related to broad cellular developmental and organismal processes, while for *TET2* CHIP the top GO terms related to cellular regulation and cell signaling (Supplementary Data 14–17). For example, for any CHIP, *DNMT3A* CHIP, and *ASXL1* CHIP, the top ten most significant ontologies included multicellular organism development, anatomical structure development, system development, and developmental process. For *TET2* CHIP, the most significant ontology terms related to a cellular response to stimulus, regulation of cellular processes, and cell signaling. Notably, for the genes annotated to the 554 *TET2*-associated CpGs that were found to be demethylated, the top GO terms were enriched for cellular developmental and organismal processes such as multicellular organism development and system development—similar to the enriched GO terms for genes annotated to *DNMT3A* CHIP-associated CpGs (Supplementary Data 29).

To understand how differentially methylated CpGs associated with CHIP might alter cellular function, we identified gene expression changes associated with CHIP-linked CpGs. We analyzed the associations of CHIP-associated CpGs with changes in *cis* gene expression (expressed gene [eGene] within 100 kB of CpG) in 2115 FHS participants whose DNA methylation data and whole-blood RNA-seq data were available. At $P < 1 \times 10^{-7}$, we identified 467 significant *cis* CpG-transcript pairs for any CHIP, 258 for *DNMT3A* CHIP, 293 for *TET2* CHIP, and 234 for *ASXL1* CHIP (Supplementary Data 18–21 provide the full expression quantitative trait methylation (eQTM) results)[19]. The vast majority of the associations between methylation and gene expression changes were negative, where decreased methylation changes were associated with increased gene expression changes or increased methylation changes were associated with decreased gene expression changes. For any CHIP, *DNMT3A*, *TET2*, and *ASXL1* CHIP, ~68% (317/467), ~71% (184/258), ~77% (224/293), and ~72% (168/234) of CpGs had a negative association between methylation and gene expression changes, respectively. For any CHIP, the top enriched GO terms related to lipid metabolism. eGenes associated with *DNMT3A* CHIP were enriched in cell motility and adhesion processes. For *TET2* CHIP, the top enriched terms related to immune processes, such as leukocyte differentiation. *ASXL1* CHIP eGenes were enriched in cellular and immune processes, including cell importation and antigen processing and presentation (Supplementary Data 22–25).

**Association of methylation with variants and MR analysis**
*Cis*-methylation quantitative trait loci (*cis*-mQTL)—genetic loci that are significantly associated with CpG methylation levels and located within 1 Mb of their associated CpG—linked 8642 CpGs associated with any CHIP and CHIP subtypes to GWAS Catalog traits/diseases[19,20]. Of the *cis*-mQTL variants, a subset were associated with clonal hematopoiesis traits, particularly myeloid clonal hematopoiesis and the number of clonal hematopoiesis mutations (Supplementary Data 26).

**Table 2 | Top 10 CHIP-associated CpGs**

| CHIP Subtype | CpG | CHR | Position | Gene | β | SE | P-value | Association with CHIP (Black participants)* | Association with CHIP (White participants)** |
|---|---|---|---|---|---|---|---|---|---|
| Any CHIP | cg23014425 | 17 | 46648525 | *HOXB3* | −0.016 | 8.7E-04 | 6.60E-79 | --- | --- |
| | cg04800503 | 17 | 46648533 | *HOXB3* | −0.028 | 1.5E-03 | 8.10E-76 | --- | --- |
| | cg07727170 | 15 | 70458214 | | −0.016 | 9.1E-04 | 5.30E-69 | --- | --- |
| | cg01966117 | 3 | 52528714 | *STAB1* | −0.034 | 1.9E-03 | 1.20E-68 | --- | --- |
| | cg19825437 | 3 | 1.69E+08 | | −0.038 | 2.2E-03 | 3.60E-68 | --- | --- |
| | cg25113462 | 2 | 2.39E+08 | *TRAF3IP1* | −0.023 | 1.3E-03 | 1.40E-64 | --- | --- |
| | cg08343644 | 16 | 57662060 | *GPR56* | −0.021 | 1.3E-03 | 1.10E-57 | --- | --- |
| | cg01521274 | 14 | 71822452 | | −0.025 | 1.5E-03 | 7.00E-57 | --- | --- |
| | cg21517792 | 14 | 1.06E+08 | *MTA1* | −0.024 | 1.5E-03 | 1.20E-55 | --- | --- |
| | cg15059065 | 19 | 17354961 | *NR2F6* | −0.04 | 2.6E-03 | 1.60E-53 | --- | --- |
| *DNMT3A* CHIP | cg04800503 | 17 | 46648533 | *HOXB3* | −0.048 | 1.8E-03 | 7.10E-150 | --- | --- |
| | cg23014425 | 17 | 46648525 | *HOXB3* | −0.026 | 1.0E-03 | 2.00E-143 | --- | --- |
| | cg25113462 | 2 | 2.39E+08 | *TRAF3IP1* | −0.038 | 1.7E-03 | 6.20E-112 | --- | --- |
| | cg03785076 | 2 | 2.42E+08 | *SNED1* | −0.052 | 2.5E-03 | 6.20E-95 | --- | --- |
| | cg23551720 | 17 | 46633726 | *HOXB3* | −0.038 | 1.9E-03 | 8.70E-91 | --- | --- |
| | cg07727170 | 15 | 70458214 | | −0.023 | 1.2E-03 | 4.70E-90 | --- | --- |
| | cg09749364 | 15 | 40384779 | *BMF* | −0.046 | 2.3E-03 | 5.10E-87 | --- | --- |
| | cg16937168 | 2 | 2.42E+08 | *SNED1* | −0.059 | 3.1E-03 | 9.20E-82 | --- | --- |
| | cg23146197 | 12 | 66271002 | *HMGA2* | −0.041 | 2.2E-03 | 5.00E-80 | --- | --- |
| | cg24400630 | 1 | 89728035 | *GBP5* | −0.046 | 2.5E-03 | 1.90E-78 | --- | --- |
| *TET2* CHIP | cg13742400 | 2 | 2.26E+08 | *DOCK10* | 0.086 | 4.5E-03 | 1.90E-82 | +++ | +++ |
| | cg19695507 | 10 | 13526193 | *BEND7* | 0.097 | 5.2E-03 | 3.30E-76 | +++ | +++ |
| | cg22562591 | 8 | 82002977 | *PAG1* | 0.057 | 3.4E-03 | 6.20E-65 | N/A | +++ |
| | cg07905808 | 6 | 30297389 | *TRIM39* | 0.068 | 4.0E-03 | 1.30E-64 | +++ | +++ |
| | cg00116699 | 2 | 2.40E+08 | *HDAC4* | 0.082 | 4.9E-03 | 3.70E-63 | +++ | +-+ |
| | cg06043201 | 8 | 28974428 | *KIF13B* | 0.085 | 5.1E-03 | 2.20E-62 | +++ | +++ |
| | cg09667606 | 6 | 1.59E+08 | *SYNJ2* | 0.077 | 4.6E-03 | 7.10E-62 | +++ | +++ |
| | cg17607231 | 2 | 2.31E+08 | *SP140* | 0.12 | 7.3E-03 | 1.70E-60 | +++ | +++ |
| | cg01133215 | 6 | 45399681 | *RUNX2* | 0.085 | 5.2E-03 | 1.70E-59 | +++ | +++ |
| | cg25463483 | 6 | 30530544 | *PRR3* | 0.064 | 4.0E-03 | 1.70E-58 | +++ | +++ |
| *ASXL1* CHIP | cg07262247 | 5 | 1.32E+08 | *PDLIM4* | −0.2 | 7.8E-03 | 2.10E-133 | | - |
| | cg17543112 | 5 | 1.32E+08 | *PDLIM4* | −0.14 | 5.9E-03 | 1.20E-117 | | - |
| | cg01305625 | 5 | 1.32E+08 | *PDLIM4* | −0.11 | 5.8E-03 | 2.70E-80 | | - |
| | cg17412560 | 2 | 95963403 | *KCNIP3* | −0.16 | 8.5E-03 | 3.90E-72 | | - |
| | cg00443981 | 17 | 58499679 | *C17orf64* | −0.17 | 9.7E-03 | 6.10E-64 | | - |
| | cg02544002 | 3 | 1.29E+08 | *PLXND1* | −0.14 | 8.6E-03 | 3.40E-57 | | - |
| | cg19529621 | 12 | 2045722 | | −0.15 | 9.3E-03 | 7.80E-54 | | - |
| | cg02341556 | 11 | 1.19E+08 | *BCL9L* | −0.14 | 9.2E-03 | 3.40E-53 | | - |
| | cg06124793 | 11 | 1939725 | *TNNT3* | −0.11 | 6.8E-03 | 5.60E-52 | | - |
| | cg10558233 | 8 | 94892613 | | −0.16 | 1.0E-02 | 1.40E-51 | | - |

The effect size (β), standard error (SE), and P-values for any CHIP, *DNMT3A* CHIP, and *TET2* CHIP were derived from fixed-effect meta-analysis. Because *ASXL1* CHIP was only available in the Framingham Heart Study (FHS) cohort, the β, SE, and P-values were derived from linear regression models. Two-sided tests were used for all analyses. P-values were adjusted for multiple comparisons using the Benjamini-Hochberg FDR method.
* "Association with CHIP": "+" or "-" represent the directions of effect for any CHIP, *DNMT3A*, or *TET2* CHIP in CHS, ARIC, and JHS, respectively; **"Association with CHIP": "+" or "-" represent the directions of effect for any CHIP, *DNMT3A*, or *TET2* CHIP in CHS, ARIC, and FHS, respectively; "N/A": CpG was not found in association with CHIP.
*CHIP* Clonal Hematopoiesis of Indeterminate Potential, *CHR* Chromosome.

Additionally, enrichment tests of CHIP-associated CpG sites with EWAS catalog traits[21] were performed across 4023 traits using a significance threshold of $1.24 \times 10^{-5}$ (0.05/4023) (Supplementary Data 27). For any CHIP, *DNMT3A* CHIP, *TET2* CHIP, and *ASXL1* CHIP, the top outcomes reflected CpG sites related to age/aging, alcohol consumption, smoking, and multiple CVD-related traits including body mass index (BMI), type II diabetes, and fasting insulin. In support of previous studies reporting *ASXL1* CHIP enrichment among smokers[22,23], 24% (1462/6078) of *ASXL1* CHIP-associated CpGs overlapped with smoking-associated CpGs.

Two-sample MR analysis of CHIP-associated CpGs (as exposures) with *cis*-mQTLs as the instrumental variables in relation to CVD-related traits and mortality (as outcomes) was performed to infer whether differential methylation at CHIP-associated CpGs may causally influence the outcomes. The significantly associated CpGs for any CHIP and for the three CHIP driver genes were tested for causal associations with 22 traits, including all-cause mortality, BMI, LDL cholesterol, hypertension, diabetes, CVD, and smoking. The top 20 CpGs and annotated genes for each trait are reported in Table 3 (Supplementary Data 28 displays the full MR results). 261 CHIP-associated, differentially

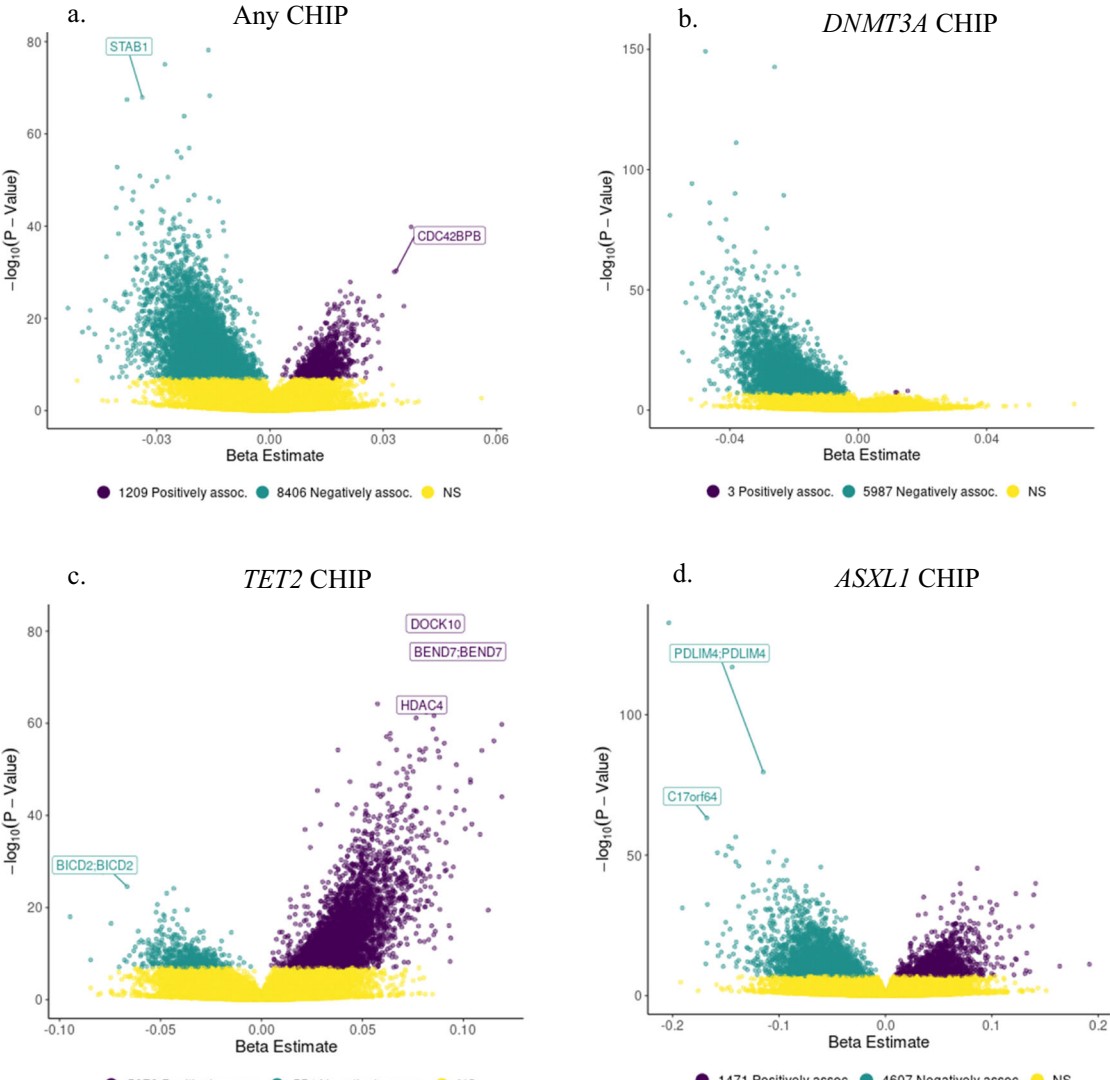

**Fig. 2 | Genome-wide Directions of Effect of Any CHIP and CHIP Subtypes.** Volcano plots showing the effect size (β) and -log₁₀(*P*-value) from the multiracial meta-analysis of epigenome-wide association studies (EWAS) for (**a**) any CHIP (Clonal Hematopoiesis of Indeterminate Potential), (**b**) *DNMT3A* CHIP, (**c**) *TET2* CHIP, and the EWAS in the Framingham Heart Study (FHS) for (**d**) *ASXL1* CHIP. Genes annotated to the CpG sites are shown. For panels (**a**–**c**), the effect size (β) and *P*-values were derived from fixed-effect meta-analysis of multiple cohorts. For panel (**d**), because *ASXL1* CHIP was only available in the FHS cohort, the effect size (β) and *P*-values were derived from linear regression models. The color green indicates a significant negative association between CHIP and DNA methylation while purple indicates a positive association between the two variables. Yellow signifies non-significant associations between CHIP and DNA methylation. Two-sided tests were used for all analyses. *P*-values were adjusted for multiple comparisons using the Benjamini-Hochberg false discovery rate (FDR) method. Significant associations were defined as FDR < 0.05. Exact *P*-values, standard errors for the β, and 95% confidence intervals for significant results are provided in Supplementary Data 7-10. Source data are provided as a Source Data file.

methylated CpG sites were identified that were putatively causally associated with CVD-related traits and/or all-cause mortality, including eight CpGs for myocardial infarction (MI) (e.g., cg11879188 (*ABO*), $\beta_{MR} = -0.99$, $P_{MR} = 4.8 \times 10^{-18}$), 108 CpGs for blood pressure (e.g., cg20305489 (*SEPT9*), $\beta_{MR} = 10$, $P_{MR} = 1.7 \times 10^{-31}$), 86 CpGs for lipid traits (e.g., cg11250194 (*FADS2*), $\beta_{MR} = -0.89$, $P_{MR} = 2.0 \times 10^{-33}$), and two CpGs for mortality (e.g., cg08756033 (*C13orf33*), $\beta_{MR} = 0.016$, $P_{MR} = 1.3 \times 10^{-4}$). 53 CpGs were associated with more than one trait. For example, cg11879188 is annotated to the *ABO* gene and was associated with seven traits, including diastolic blood pressure ($\beta_{MR} = 2.7$, $P_{MR} = 1.9 \times 10^{-23}$), MI ($\beta_{MR} = -0.99$, $P_{MR} = 4.8 \times 10^{-18}$), and triglycerides ($\beta_{MR} = 0.20$, $P_{MR} = 2.0 \times 10^{-9}$).

## Discussion

We report the results of a multiracial meta-EWAS of CHIP and identified thousands of CpG sites across the genome that are significantly associated with any CHIP and with *DNMT3A*, *TET2*, and *ASXL1* CHIP. Of note, the vast majority of the CpGs were *trans*- relative to the CHIP driver gene. This appears to be consistent with the functions of *DNMT3A*, *TET2*, and *ASXL1* in globally altering DNA methylation levels of CpG sites genome wide, as seen in the EWAS of each of the three CHIP driver genes, where the significantly associated CpGs were numerous and located diffusely across the genome. The methylomic signatures of CHIP and CHIP driver genes were experimentally validated with human-engineered CHIP cells. Downstream analyses were conducted to assess whether these alterations in DNA methylation levels may be causally associated with CVD-related outcomes and all-cause mortality. Causal inference analyses using two-sample MR revealed evidence of a possible causal role of CHIP-associated CpGs in various CVD-related traits and all-cause mortality.

For the experimental validation of our meta-EWAS results, any CHIP-associated CpG sites were significantly enriched in *DNMT3A*-

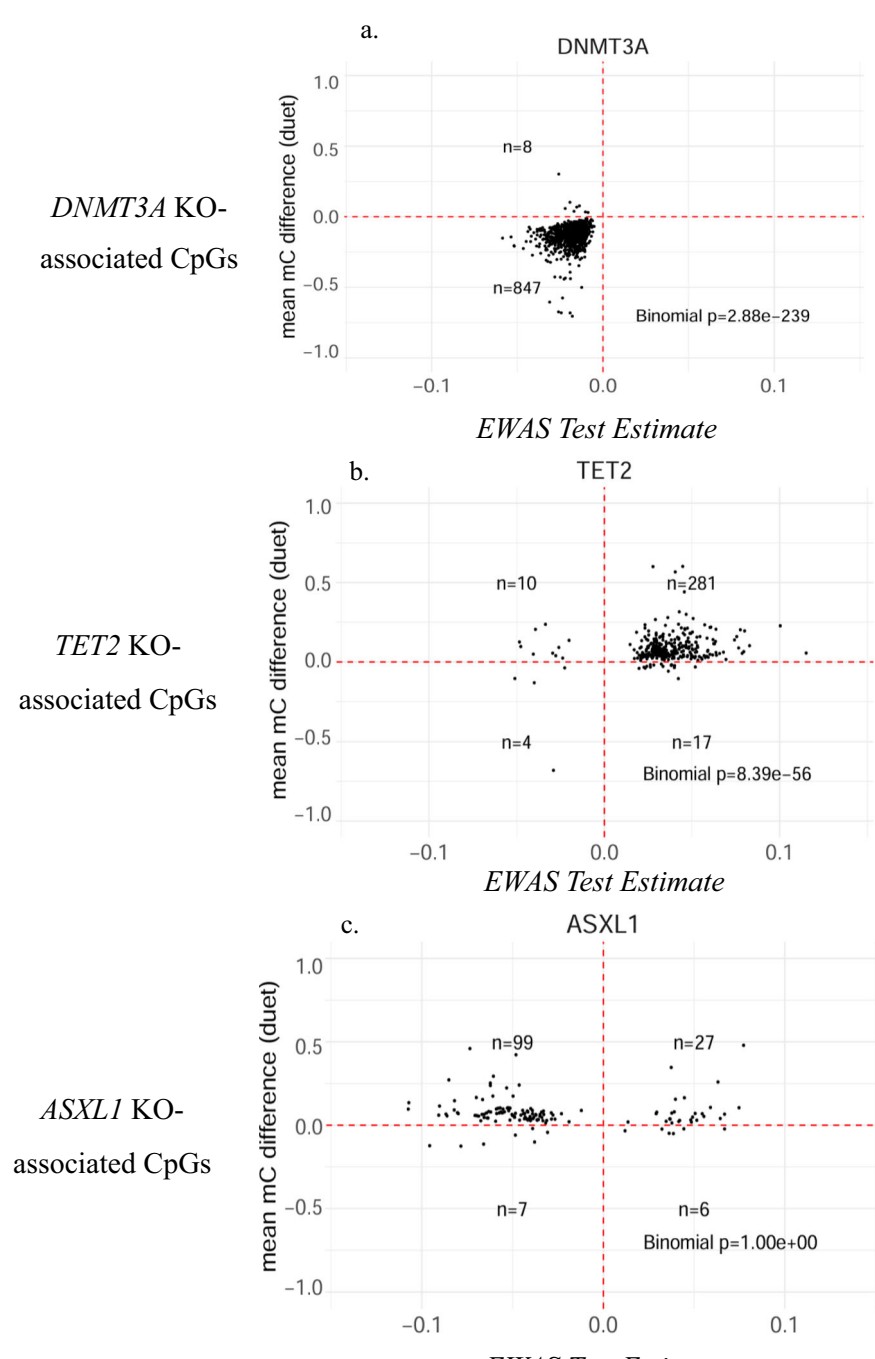

**Fig. 3 | Functional Validation in CRISPR/Cas9-edited Hematopoietic Stem Cells Modeling CHIP.** Dot plots of methylation change from −1.0 (no methylation) to 1.0 (complete methylation) seen in engineered primary cell cultures compared to correlation of EWAS results ranging from −0.1 to 0.1. Following an initial CpG filtering using an uncorrected Student's *t*-test (*p* < 0.05), significance was determined with a two-sided binomial test. **a** *DNMT3A*-associated CpG sites (*n* = 855 CpG sites) compared to *DNMT3A* engineered human stem cells (*n* = 4). **b** *TET2*-associated CpG sites (*n* = 312) compared to *TET2*-engineered human stem cells (*n* = 4). **c** *ASXL1*-associated CpG Sites (*n* = 139) compared to *ASXL1*-engineered human stem cells (*n* = 3). Source data are provided as a Source Data file. *KO* Knockout, *mC* methylcytosine.

engineered cells, which was expected given the overwhelming predominance of *DNMT3A* CHIP among total CHIP cases reported in our study and several others[2,5,12]. Interestingly, *TET2*-associated CpG sites were enriched in *ASXL1*-engineered cells. This finding is consistent with the substantial CpG overlap (~1000 shared CpGs) between *TET2* and *ASXL1* CHIP from the meta-EWAS and suggests that the epigenetic regulators *TET2* and *ASXL1* impact several of the same genome regions and may lead to similar downstream consequences. Notably, *ASXL1*-associated CpGs showed no significant enrichment in the *ASXL1*-

engineered cells. The lack of enrichment of *ASXL1*-associated CpGs in the *ASXL1*-engineered cells may limit the validity of the study's downstream analyses with *ASXL1* CHIP. This observation may be due to the limited number of *ASXL1* CHIP cases in the EWAS as well as several biological factors. *ASXL1* mutations primarily affect histone modifications, particularly H2AK119 ubiquitination, which indirectly influences chromatin accessibility[24]. Recent studies have shown that *ASXL1* loss-of-function mutations increase chromatin accessibility, potentially resulting in individualistic methylation changes influenced by other

**Table 3 | Mendelian Randomization of CHIP-associated CpGs and CVD-related Outcomes**

| CHIP Category | Exposure | Outcome | β | SE | *P*-value | FDR | Nearby Gene | Association with CHIP* |
|---|---|---|---|---|---|---|---|---|
| *DNMT3A* | cg11250194 | LDL cholesterol | −0.89 | 0.074 | 2.0E-33 | 1.1E-28 | *FADS2* | - |
| *TET2* | cg20305489 | Diastolic blood pressure | 10 | 0.89 | 1.7E-31 | 6.7E-27 | *SEPT9* | + |
| *TET2* | cg20305489 | Systolic blood pressure | 16 | 1.5 | 3.6E-26 | 1.3E-21 | *SEPT9* | + |
| Any CHIP/ *DNMT3A/ASXL1* | cg00776080 | Diastolic blood pressure | −28 | 2.7 | 2.6E-24 | 7.9E-20 | *TENC1* | - |
| Any CHIP/*ASXL1* | cg11879188 | Diastolic blood pressure | 2.7 | 0.27 | 1.9E-23 | 4.7E-19 | *ABO* | - |
| Any CHIP | cg24530246 | HDL cholesterol | 0.46 | 0.047 | 1.2E-22 | 2.7E-18 | | - |
| *DNMT3A* | cg11250194 | HDL cholesterol | −1.5 | 0.16 | 1.3E-22 | 2.7E-18 | *FADS2* | - |
| *DNMT3A* | cg11250194 | Triglycerides | 1.4 | 0.14 | 1.8E-22 | 3.6E-18 | *FADS2* | - |
| *DNMT3A* | cg16517298 | HDL cholesterol | 0.56 | 0.062 | 1.1E-19 | 1.6E-15 | *GALNT2* | - |
| Any CHIP/*DNMT3A* | cg17892169 | Diastolic blood pressure | 8.9 | 0.99 | 2.2E-19 | 3.2E-15 | *TNFSF12* | - |
| *TET2* | cg01687878 | Diastolic blood pressure | −6.7 | 0.75 | 3.8E-19 | 5.2E-15 | | + |
| *TET2* | cg10632966 | Systolic blood pressure | 25 | 2.8 | 3.3E-18 | 3.8E-14 | | + |
| Any CHIP/*ASXL1* | cg11879188 | Myocardial infarction | −0.99 | 0.11 | 4.8E-18 | 5.3E-14 | *ABO* | - |
| *DNMT3A* | cg16517298 | Triglycerides | −0.55 | 0.064 | 8.0E-18 | 7.9E-14 | *GALNT2* | - |
| Any CHIP | cg00417151 | HDL cholesterol | 0.85 | 0.10 | 2.7E-17 | 2.6E-13 | *RRBP1* | - |
| *TET2* | cg16060189 | Type 2 diabetes | 2.4 | 0.29 | 1.1E-16 | 9.4E-13 | | + |
| *TET2* | cg14016363 | Diastolic blood pressure | −21 | 2.5 | 1.3E-16 | 1.0E-12 | | + |
| Any CHIP | cg00526336 | Triglycerides | −2.1 | 0.28 | 2.5E-14 | 1.4E-10 | *GALNT2* | - |
| Any CHIP/*DNMT3A* | cg06346307 | Systolic blood pressure | −11 | 1.5 | 2.7E-14 | 1.5E-10 | *COMT* | - |
| Any CHIP | cg19758448 | HDL cholesterol | −0.54 | 0.073 | 1.0E-13 | 5.5E-10 | *PGAP3* | - |

The effect size (β), standard error (SE), and *P*-values were derived from two-sample Mendelian randomization (MR) tests. Two-sided tests were used for all analyses. FDR values were calculated using the Benjamini-Hochberg FDR method.
*"Association with CHIP": "+" or "-" represent the directions of effect for any CHIP, *DNMT3A*, *TET2*, or *ASXL1* CHIP in meta-EWAS.
*CHIP* Clonal Hematopoiesis of Indeterminate Potential.

genetic and environmental factors[25]. Furthermore, the effects of *ASXL1* mutations on methylation might be temporally dynamic or cell-type specific, aspects not fully captured in our current experimental design. Future studies with larger sample sizes, particularly for *ASXL1* CHIP, and longer observation periods may help elucidate the relationship between *ASXL1* mutations and DNA methylation patterns in CHIP and determine with greater certainty the epigenetic signatures of this CHIP driver mutation.

By incorporating an engineered reductionist system, we provide an orthogonal approach to confirm that the patterns observed in CHIP donors result directly from the somatic mutation. By engineering *DNMT3A*, *TET2*, and *ASXL1* mutations into healthy CD34+ cells and performing DNA methylation profiling, we can recapitulate the DNA methylation patterns seen in CHIP donors. Our study also establishes this reductionist system as a robust representation of the methylation phenotype. Future studies leveraging this system will enable more precise dissection of the causal relations between these mutations and changes in DNA methylation than would be possible from population-scale epidemiology data alone.

In the in vitro validation analysis, we focused on the subset of CpG sites that were significantly associated with CHIP in the EWAS and nominally differentially methylated in the in vitro CHIP model to improve the validity of our findings and reduce the likelihood of false positives. By concentrating on overlapping CpG sites, we prioritized CpGs with a stronger potential biological relevance, as they were consistent across population-level and experimental settings. Importantly, despite the strong directional concordance in the subset of

overlapping CpG sites, a small proportion of the CpGs identified from the meta-EWAS were captured in the in vitro model: ~14% (855/5990) for *DNMT3A*, ~6% (312/5633) for *TET2*, and ~2% (139/6078) for *ASXL1* CHIP. This may be because the in vitro system does not fully capture the complex in vivo environment in which CHIP is influenced by various cell types, environmental factors, and systemic interactions, such as immune system interactions.

Two-sample MR analysis identified 261 differentially methylated CpG sites that were putatively causally related to one or more CVD traits and/or all-cause mortality. For example, cg11250194 was putatively causally associated with four CVD-related cardiometabolic traits: LDL cholesterol, HDL cholesterol, triglycerides, and fasting glucose. Cg11250194 resides in the *FADS2* gene. It is hypomethylated, associated with *DNMT3A* CHIP (β = −0.022, *P* = 1.6E-13), and replicated in the *DNMT3A* CHIP-engineered cells. The *FADS2* gene encodes the enzyme fatty acid desaturase 2 – the first rate-limiting enzyme for the biosynthesis of polyunsaturated fatty acids[26]. A recent study found that cg11250194 (*FADS2*) was associated with Alternative Healthy Eating Index and that hypermethylation of this CpG was associated with lower triglyceride levels[27]. Moreover, cg11250194 was previously identified in an EWAS of lipid-related metabolic measures[28]. Based on our findings, hypomethylation of this diet-associated CpG may be linked to higher triglyceride levels, putatively increasing the risk for CVD. *FADS2* overexpression has also been found to promote clonal formation[26]. Thus, *FADS2* may be an important gene connecting CHIP with diet. Of note, of the 30 CpGs associated with either Mediterranean-style Diet Score or Alternative Healthy Eating Index or

both in a 2020 study by Ma et al.[27], 17 were CHIP-associated CpGs (~57%) identified from our multiracial meta-EWAS of CHIP. The substantial overlap between diet- and CHIP-associated CpGs is consistent with the hypothesis that an unhealthy diet may be associated with CHIP through epigenetic mechanisms. Despite including smoking status as a covariate in the models for all participating cohorts in both studies[28], we recognize that our ability to fully separate the effects of diet from smoking on CHIP risk may be limited due to our cross-sectional study design. As a result, some of the observed association between unhealthy diet and CHIP through epigenetic mechanisms may reflect residual smoking effects.

Compared to a previously published EWAS of CHIP[12] ($N = 3273$, 228 CHIP cases), the present study has a substantially larger sample size ($N = 8196$, 462 CHIP cases), including all the samples from the previous study. With the larger sample size of the present study, we identified 6687, 3524, and 4678 novel CpGs significantly associated with any CHIP and with the top two CHIP driver genes *DNMT3A* and *TET2*. Of the CpG sites identified from the previous EWAS study at $P < 1 \times 10^{-7}$, a large proportion overlapped and have concordant directions of effect with CpGs from the multiracial meta-EWAS of CHIP at $P < 1 \times 10^{-7}$: 91% (2928/3217) for any CHIP, 89% (2466/2769) for *DNMT3A* CHIP, and 90% (955/1059) for *TET2* CHIP. This is expected, as almost half of the CHIP cases in our meta-EWAS of CHIP are from the previous EWAS of CHIP[12]. Additionally, we report thousands of *ASXL1* CHIP-associated CpGs from an EWAS of *ASXL1* CHIP. Through eQTM analysis that identified CpG-transcript pairs, the top eGenes in *ASXL1* CHIP relate to various immune processes, suggesting that dysregulated immune function may contribute to *ASXL1* CHIP-related disease outcomes. This putative role of *ASXL1* CHIP in perturbing immune function, specifically T cell function, has been recently reported using an *ASXL1* CHIP conditional knock-in mouse model[29]. Notably, several of the *ASXL1* CHIP-associated CpGs displayed putatively causal relations to CVD-related traits in MR analysis, including cg11879188 (in *ABO*).

While there are several strengths of our study, some limitations should be noted. Although smoking status was included as a covariate in the statistical models for all study cohorts, there could be residual confounding as smoking behavior may not be fully adjusted for in the analysis. Thus, smoking could still be driving the association between CHIP and CVD, as was reported in a recent study of CH[30]. Of note, there are several studies across diverse populations and in different settings that controlled for smoking as a covariate and also found an association of CHIP (a specific form of CH) with CVD. For instance, in a recent study, Diez-Diez et al.[31] clarified the directionality of the CHIP-CVD relationship with adjustment for smoking and concluded that CH confers an increased risk of developing atherosclerosis.

Additionally, the way that the DeCODE[30] investigators ascertained "CH" may have contributed to their finding that CH was not associated with CVD, as it is distinct from the definition of "CHIP." CH includes clonal events with known leukemic driver gene mutations, such as CHIP and mosaic chromosomal alterations (mCAs), and clonal events without clear driver genes. Given recent findings that each of these distinct classes of CH has unique phenotypic consequences[2,32], the lack of association between CH and CHIP reported by Stacey et al. may be due to the grouping of heterogenous CH subtypes.

Moreover, it has been previously demonstrated that small changes in the stringency with which CHIP is ascertained can have an outsize effect on downstream analyses. For example, Vlasschaert et al.[33] reported the importance of CHIP detection stringency in relation to CHIP-associated CVD risk. Specifically, more stringent criteria (≥5 supporting reads) were associated with CVD risk, while less stringent criteria (≥3 supporting reads) attenuated the association of CHIP with CVD. Their study provides an up-to-date and nuanced explanation of the CHIP-CVD relationship.

Driverless CH is the occurrence of clonal expansions in blood without a known CHIP driver mutation and is estimated to drive the majority of clonal expansions in the elderly[34]. Bernstein et al.[34] identified regions within exome sequences that are under positive selection to identify additional driver mutations in whole blood (large clones >0.1) together with validation of positive selection in single cell-derived hematopoietic myeloid and lymphoid colonies. The inclusion of mutations in these fitness-inferred CH genes increases prevalence of CH by 18% in the UK Biobank cohort. In our study, CHIP was defined when an individual harbored at least one deleterious insertion/deletion or single nucleotide variant in any of the 74 genes that have been previously linked to myeloid malignancy at a variant allele frequency of at least 2%. Given the study's scope, we did not include driverless CH in our CHIP definition. Thus, CHIP prevalence in our study may be underestimated relative to studies that account for driverless clonal expansion.

A larger sample size is needed to examine less frequently mutated CHIP driver genes, such as *TP53*, *JAK2*, and *PPM1D*. Moreover, the reported putatively causal associations of CpGs with CVD outcomes and mortality were based on two-sample MR analysis. Despite our attempt to minimize horizontal pleiotropy by excluding *cis*-mQTLs that also serve as *trans*-mQTLs and excluding CpGs with three or more independent instrumental variables through MR-Egger using a threshold of $P$-value < 0.05, the two-sample MR approach has known limitations[35]. Methods for detecting and addressing pleiotropy may be ineffective[36] and, thus, longitudinal and functional studies are needed to reinforce causal findings.

To account for the possibility that the VAF of CHIP driver gene mutation may influence DNA methylation, we used a threshold of FDR < 0.05 to detect associations between CpGs and VAF. There were no significant associations between CpG sites and VAF. Based on these results, we do not believe that VAF significantly influenced our DNA methylation findings. Notably, the sample size with available VAF information is limited. For example, in the FHS cohort, we have only 166 CHIP cases and cannot completely rule out the possibility that VAF may still have a modest impact on DNA methylation. Experimental studies or studies with larger sample sizes may be necessary to address the effect of VAF of mutation on DNA methylation.

While this study benefits from a large sample size, which allows for robust statistical comparisons, we acknowledge the distinction between statistical significance and biological significance. The statistical significance observed in this study does not necessarily equate to meaningful biological effects. The impact of cell type correction on downstream analyses, particularly in heterogeneous patient populations, has not been fully validated. Further research is needed to determine how this adjustment translates into biological outcomes. Additionally, our study evaluated individual CpG sites, however, differentially methylated regions consisting of several consecutive methylated CpGs have been shown to have important implications for disease pathogenesis[37]. Thus, studies exploring these broader methylation patterns are warranted to better capture the functional relevance of epigenetic signatures of CHIP.

Last, although cell-type proportions were included as covariates for all cohorts, we cannot exclude the possibility that subtle uncorrected effects in cell-type proportions due to clonal selection in immune cells may contribute to the enrichment of immune function observed for *TET2* and *ASXL1* CHIP eGenes. While cell-type adjustments reduce confounding effects, residual contributions from altered immune cell proportions remain possible. Future studies investigating cell-type specific DNA methylation and gene expression may provide additional clarity on the impact of CHIP on immune gene expression.

Overall, our study sheds light on the epigenetic changes linked to CHIP and CHIP subtypes and their associations with CVD-related outcomes. The differentially expressed genes and pathways linked to the epigenetic features of CHIP may serve as therapeutic targets for CHIP-related diseases. For example, *Fc receptor-like protein 3 (FCRL3)* (cg17134153, $Fx = -5.5$, $P = 1E-113$) is the top differentially expressed gene for *TET2* CHIP. *FCRL3* encodes a type I transmembrane

glycoprotein that is expressed by lymphocytes and plays a role in modulating immune responses[38]. Polymorphisms in this gene have been implicated in the pathogenesis of autoimmune diseases[38,39]. A recent study demonstrated that FCRL3 stimulation of regulatory T cells induced production of pro-inflammatory cytokines, including IL-17 and IL-26[38]. This finding suggests that FCRL3 may play a critical role in mediating the transition of regulatory T cells to a pro-inflammatory phenotype and could potentially contribute to the increased inflammation observed among *TET2* CHIP carriers[40,41]. Additionally, Clark et al. identified *FCRL3* as a gene for which DNA methylation at the CpG site cg17134153 in CD4$^+$ T cells likely mediates the genetic risk for rheumatoid arthritis[42]. Given that CHIP, including *TET2* CHIP, has been associated with rheumatoid arthritis (RA)[43], the regulation of *FCRL3* expression by methylation changes at cg17134153 may, in part, serve as the functional basis of the observed association between CHIP and RA. Further experimental studies are warranted to better understand how differential expression of *FCRL3* may impact *TET2* CHIP development and the pathogenesis of RA. Taken together, our results provide insight into the molecular mechanisms underlying age-related diseases, namely cardiovascular disease.

## Methods

### Ethics

All participants provided written, informed consent. The study protocol was approved by the following institutional review boards at each collaborating institution: Institutional Review Board at Boston Medical Center (FHS); University of Washington Institutional Review Board (CHS); University of Mississippi Medical Center Institutional Review Board (ARIC: Jackson Field Center); Wake Forest University Health Sciences Institutional Review Board (ARIC: Forsyth County Field Center); University of Minnesota Institutional Review Board (ARIC: Minnesota Field Center); Johns Hopkins University School of Public Health Institutional Review Board (ARIC: Washington County Field Center); University of Mississippi Medical Center (JHS); Jackson State University (JHS); and Tougaloo College (JHS). All research was performed in accordance with relevant ethical guidelines and regulations. The study design and conduct adhered to all relevant regulations regarding the use of human study participants and was conducted in accordance to the criteria set by the Declaration of Helsinki.

### Study cohorts

The Framingham Heart Study (FHS) is a prospective, observational community-based cohort investigating risk factors for CVD. For our discovery sample, DNAm was measured from FHS participants ($N = 3295$) in the Offspring cohort ($N = 1860$; Exam 8; years 2005-2008)[37] and in the Third Generation cohort ($N = 1435$; Exam 2; years 2008-2011)[44]. CHIP calls were based on whole-genome sequencing of whole blood DNA samples, the majority of which were from FHS Offspring participants at Exam 8 and Gen 3 participants at Exam 2 and temporally concordant with the time of DNAm profiling. All FHS participants self-identified as White at the time of recruitment.

The Jackson Heart Study (JHS) is an observational community-based cohort studying the environmental and genetic factors associated with CVD in African Americans. For our discovery sample, data were collected from 1664 JHS participants[12]. DNAm was measured from the majority of JHS participants at visit 1, with a small subset at visit 2. CHIP calls were concurrent with DNAm profiling and based on whole-genome sequencing of whole blood DNA samples, where the majority were from visit 1 (years 2000–2004) and a subset from visit 2 (years 2005-2008)[12]. All JHS participants self-identified as Black or African American at the time of recruitment. No ancestry outliers were excluded, as inferred based on genetic similarity to reference panels. Similarity to the 1000 G AFR reference panel varied by individual (study q1, median, q3 77.9% 84.3% 89.0%) in the methylation and WGS overlap dataset, using estimates from RFMix.

The Cardiovascular Health Study (CHS) is a population-based cohort study of risk factors for CVD in adults aged 65 or older[45]. DNAm was measured from blood samples from participants in years 5 and 9, year 5, or year 9 only. CHIP calls were based on whole-genome sequencing of blood samples, where the majority were taken 3 years before or concurrently with the first DNAm measurement[12]. CHS participants self-reported their race at the time of recruitment.

The Atherosclerosis Risk in Communities (ARIC) is a prospective, multiracial cohort study of risk factor and clinical outcomes of atherosclerosis[38]. DNAm was measured from 2655 ARIC participants at visit 2 (1990-1992) or visit 3 (1993-1995). CHIP calls were based on whole exome sequencing of blood samples from visit 2 and visit 3[12,39]. ARIC participants self-identified their race at the time of recruitment. There is a subset of participants included in both ARIC and JHS. These overlapping participants were not excluded.

### DNA methylation profiling

All the DNA samples were from whole blood. The four cohorts including FHS, JHS, CHS and ARIC, conducted independent laboratory DNAm measurements, quality control (including sample-wise and probe-wide filtering and probe intensity background correction; see Supplementary Information File). DNA methylation was measured in FHS, CHS, and ARIC participants using Illumina Infinium Human Methylation-450 Beadchip (450 K array) and in JHS participants using the Ilumina EPIC array[40,41].

### CHIP calling

For the purposes of this investigation, CHIP was defined as a candidate driver gene mutation in genes that have been reported to be associated with hematologic malignancy, is present at a variant allele frequency (VAF) of at least 2% in peripheral blood, and is present in the absence of hematologic malignancy[42]. CHIP was detected in FHS, JHS, and CHS from WGS blood DNA in the NHLBI Trans-Omics for Precision Medicine (TOPMed) consortium using the Mutect2 software[5]. In ARIC, CHIP calls were based on whole exome sequencing of blood DNA using the same procedure[5]. CHIP is defined as when an individual harbors at least one pre-specified deleterious insertion/deletion or single nucleotide variant in any of the 74 genes linked to myeloid malignancy at a variant allele frequency (VAF) $\geq 2\%$[5]. TOPMed WGS samples were sequenced to a median depth of 40x, with the sequencing depth ranging from 30x-50x for a specific region. At this sequencing depth, CHIP can be reliably ascertained with a VAF > 10% but CHIP variants with a VAF $\leq$ 10% are unable to be robustly captured[5]. For a sensitivity analysis, race-stratified and multiracial meta-EWAS of any CHIP was performed using a more restrictive CHIP clone size of VAF > 10% (See Supplementary Fig. 4 and Supplementary Data 11-13).

### Cohort-specific EWAS

The correction of methylation data for technical covariates was cohort specific. Each cohort performed an independent investigation to select an optimized set of technical covariates (e.g., batch, plate, chip, row, and column), using measured or imputed blood cell type fractions, surrogate variables, and/or principal components. Most cohorts had previous publications using the same dataset for EWAS of different traits, such as EWAS of alcohol drinking and smoking. In this study, those cohorts used the same strategies as they did previously for correcting for technical variables, including batch effects. Linear mixed models were used to test the associations between CHIP status as the predictor variable and DNAm $\beta$ values as the outcome variable. Information about cohort-specific models is available in the Supplementary Information File.

### Meta-analysis

All analyses were contingent on self-reported Black or White race. Previous ancestry inference in these cohort studies[43] suggests high

genetic similarity of nearly all self-identified White participants to EUR reference panels (including 1000 Genomes). Self-identified Black participants have high but variable (average ~80% but may vary based on study and by study participant) genetic similarity to AFR reference panels and have some similarity to EUR reference panels as well. In some cases, extreme ancestry outliers may have been removed during study-specific QC. However, this has not been thoroughly documented in the data we received from participating studies. Importantly, we do not mean to imply that socially constructed racial identities reported by study participants are synonymous with genetic ancestry. Stratification by race may, however, capture differential social and environmental exposures within the US, which may impact the epigenome.

The meta-analysis was performed for any CHIP, *DNMT3A*, and *TET2* in White participants from FHS, CHS, and ARIC ($n = 4355$) and Black participants from JHS, CHS, and ARIC ($n = 3841$) participants, respectively, using inverse variance-weighted fixed-effects models implemented in *metagen()* function in R packages (https://rdrr.io/cran/meta/man/metagen.html). The summary statistics were used from the previous EWAS of CHIP for the ARIC and CHS cohorts[12]. Then, multiracial meta-analysis was performed for White and Black participants ($n = 8,196$). The meta-analysis was constrained to methylation probes passing filtering criteria in all cohorts.

Supplementary Fig. 1 presents QQ plots with genomic control (GC) inflation factor ($\lambda$) to illustrate the EWAS results in each cohort and in the meta-analysis. Our observations reveal a prevalence of high inflation factors ($\lambda > 1.1$) across nearly all studies. Such elevated inflation factors typically signal potential bias in the analysis process. However, it's important to note that in cases where a significant portion of CpG sites exhibit differential methylation associated with the outcome (e.g., age and CHIP), this can contribute to the observed high $\lambda$ values. Moreover, adjusting for additional PCs moderately associated with the outcome may alleviate lambda values, albeit at the expense of reduced power to detect CpGs related to the outcome. To address this, we adopted strategies consistent with those employed by the respective cohorts in previous analyses, focusing on correcting for technical variables and latent factors identified in prior studies across multiple outcomes[46–48]. Furthermore, prior to meta-analysis, we implemented additional corrections for individual study results exhibiting $\lambda > 1.5$, ensuring the integrity of our findings. The statistical significance threshold was $P < 0.05/400,000 \approx 1 \times 10^{-7}$. A less stringent threshold, the Benjamini-corrected FDR adjusted $p$-value $< 0.05$, was also used.

### Expression quantitative trait methylation analysis
Association tests of DNAm and gene expression were previously performed in 2115 FHS participants in the Offspring ($n = 686$) and Third Generation ($n = 1429$) cohorts with available whole blood DNA methylation and RNA-seq gene expression data to identify CpG sites at which differential methylation is associated with gene expression[49]. Approximately 70,000 significant *cis* CpG-transcript pairs were identified at $P < 1 \times 10^{-7}$. *Cis* is defined as CpGs located within 100 kB of the transcription start site of a mRNA. When calculating the association between CpG sites and gene-level transcripts, linear regression models were used. Residualized gene expression served as the outcome and residualized DNA methylation β value as the primary explanatory variable, with adjustment for age, sex, white blood cell count, blood cell fraction, platelet count, five gene expression PCs, and ten DNA methylation PCs. Through integration of CpGs and gene-level transcripts (mRNAs) from RNA-seq, mRNAs were identified that were significantly associated with each of the CpGs in *cis* for any CHIP and the CHIP subtypes[49,50].

### Pathway enrichment analysis
Enrichment analysis for CHIP EWAS signatures with a significance threshold of $P < 1 \times 10^{-7}$ was conducted on gene sets comprising genes annotated to CpGs associated with CHIP and major CHIP subtypes using missMethyl R package. This package adjusts for known DNAm

array bias[51]. For the enrichment analysis for eQTM gene sets, the DAVID Bioinformatics online tool was used (https://david.ncifcrf.gov/home.jsp). To improve the focus of this study, only the results of Gene Ontology (GO) terms related to biological process and KEGG pathways were used. Over-representation enrichment tests, specifically one-sided Fisher's exact tests, were used to assess whether a GO/KEGG term is significantly enriched compared to the background. The significant threshold of FDR adjusted $p$-value $< 0.05$ was used, corrected by multiple tested terms[5].

### Cell culture of mPB CD34+ cells
Patients were given G-CSF $\leq 10$ mcg/kg/day for up to 5 days. Peripheral blood mononuclear cells were collected and CD34+ cells were isolated using a MACs sorter. Samples were then counted and frozen down for future use. This research is funded from NIDDK. Mobilized peripheral blood (mPB) CD34+ cells were bought from StemCell technologies or the Cooperative Center of Excellence in Hematology (CCEH) at the Fred Hutch Cancer Research Center, Seattle, USA. The name and source of all cell lines used are the following: mPB-001: Sex: Female, Supplier Fred Hutchinson; mPB-002: Sex: Male, Supplier Fred Hutchinson; mPB-003: Sex: Female, Supplier StemCell Technologies; mPB-004: Sex: Male, Supplier: StemCell Technologies; mPB-005: Sex: Male, Supplier: StemCell Technologies. CD34+ cells were thawed and cultured in CD34+ expansion medium (StemSpan II (StemCell Technologies) + 10% CD34+ expansion supplement (Stemcell Technologies) + 20 U/mL penicillin-streptomycin (Gibco) + 500 nM UM729 (StemCell Technologies) + 750 nM Stemreginin-1 (StemCell Technologies)) for 48 h prior to editing with CRISPR-Cas9. After 48 h, samples were electroporated with RNP complexes and seeded at 400k cells per mL. Cells were maintained between 200k - 1M cells per mL.

### CRISPR-Cas9 of mPB CD34+ cells
Ribonucleoprotein (RNP) complexes targeting scramble, AAVS, *TET2*, ASXL-1, and DNMT3A were made by incubating Cas9 (IDT Alt-R HiFi sp Cas9 Nuclease V3) and sgRNA (IDT Alt-R Cas9 sgRNAs) at a 1:3.26 ratio. Guides for each gene are present in Supplementary Table 1. On day 2 post thaw, mPB CD34 cells were counted and resuspended in Buffer R or GE Buffer. RNP complexes and cells were mixed and electroporated using Neon Pipette (Thermo Scientific Inc.) with the following settings: 1650 V 10 ms pulses 3 times. Samples were seeded in expansion media at 400k/mL.

### Assessment of indel formation
Genomic DNA (gDNA) was isolated and amplified with the following conditions: 95 °C for 2 min followed by 35 cycles of 95 °C for 45 s, 61-62 °C for 1 min, 72 °C for 2 min with a final extension at 72 °C for 5 min using primers towards *TET2, ASXL-1, and DNMT3A* (Supplementary Table 2). PCR products were sent to GeneWiz (Azenta Life Sciences) where PCR cleanup and Sanger sequencing was performed. Indel formation was assessed using TIDE (Supplementary Table 3)[52].

### FACS sorting of mPB CD34+ cells
Edited CD34+ cells were sorted at day 7 post CRISPR-Cas9 using a FACSymphony™ S6 Cell Sorter or a BD FACS Aria II to remove differentiated cells. Briefly, CD34+ cells were washed in cell staining buffer (Biolegend) once and stained with antibodies targeting CD34 (Biolegend: 343614; dilution: 1:50), CD38 (Biolegend: 303532; dilution: 1:100), and Lineage Markers (Biolegend: 348805; dilution: 1:10) for 30 min at 4 °C in the dark (Supplementary Fig. 6). The antibodies are present in Supplementary Table 4.

### Duet evoC library generation and primary methylation analysis
DNA was extracted using Micro kits (Qiagen) from flow sorted cells from 3–5 donors. EvoC libraries were created following manufacturer

instructions (Biomodal). Briefly, DNA was sheared using a Covaris LE220 and assessment of input DNA was performed using Bioanalyzer instrument (Agilent) and Qbit (ThermoFisher). Library generation was performed according to the duet evoC library generation protocol (biomodal).

## Sequencing of duet evoC libraries
Capture of CpG sites was performed using Twist Human Methylome Panel (Twist Biosciences) and next generation sequencing was completed by using the NovaSeq 6000 (150 bp paired-end reads) targeting 160 M reads per sample. Biomodal pipeline version 1.1.1 was used to analyze the raw FASTQs with default settings. Briefly adapter trimming was performed with *cutadapt*, resolution of R1 and R2 to generate single-end reads with epigenetic information, mapping onto the human genome (GRCh38), and quantification of the modification state of each CpG site.

## Comparisons between EWAS and biomodal data
For each sample and for each CpG, read counts from the forward and reverse strand were summed and the mC fraction calculated as the number of reads supporting mC divided by the total number of reads with modified or unmodified C (excluding reads with A, T or G). The dataset was reduced to the CpGs with significant levels of association from each EWAS analysis. For each of these CpGs, methylation difference was calculated as the difference between the average mC fraction of multiple replicates of different KO primary cells ("DNMT3A", "TET2", "ASXL-1") and the average mC fraction of multiple replicates of control cells ("Scramble" or "AAVS"). Only CpGs with uncorrected *p*-values < 0.05 (*t*-test) were carried forward. For each EWAS analysis ("any-CHIP", "DNMT3A_chip", "TET2_chip", "ASXL1_chip") and for each gene-KO ("DNMT3A", "TET2", "ASXL-1"), the mC fraction of these CpGs was plotted against the EWAS TE, and a binomial test was used to check for enrichment in the top-right and bottom-left quadrant indicating a sign correlation between the mC fraction change induced by the KO and the EWAS TE.

## Cis-mQTLs
Methylation quantitative trait loci (mQTLs) – SNPs associated with DNA methylation – were identified from 4,170 FHS participants as previously reported[40], including 4.7 million *cis*-mQTLs at $P < 2 \times 10^{-11}$. Genotypes were imputed using the 1000 Genomes Project panel phase 3 using MACH / Minimac software. SNPs with MAF > 0.01 and imputation quality ratio >0.3 were retained. *Cis*-mQTLs were defined as SNPs residing within 1 Mb upstream or downstream of a CpG site.

## Association of methylation with complex diseases and traits
To annotate CHIP-associated CpGs and *cis*-mQTLs, we utilized both the EWAS Catalog (https://www.ewascatalog.org/)[22] and the GWAS Catalog (https://www.ebi.ac.uk/gwas/)[20]. The EWAS Catalog collected published CpG signatures for about 4000 traits and/or diseases. GWAS Catalog collected significant SNPs associated with thousands of traits and/or diseases. For the identified CHIP-associated CpGs, we matched these CpGs with reported trait-associated CpGs in the EWAS Catalog. To evaluate the enrichment of CHIP-associated CpGs for traits listed in the EWAS Catalog, we performed one-sided Fisher's exact tests. We applied a Bonferroni-corrected significance threshold of $P = 1.24E-05$ (0.05/4023, accounting for 4023 traits in the EWAS Catalog). Additionally, to assess whether any *cis*-mQTLs of CHIP-associated CpGs demonstrated strong associations with human complex traits, we matched the *cis*-mQTLs against SNPs in the GWAS Catalog that were reported with $P < 5E-8$.

## Mendelian randomization analysis
In order to investigate whether differentiation methylation at CHIP-associated CpGs causally influences risk of CVD and mortality, two-sample Mendelian randomization (MR) was performed between

exposures (CHIP-associated CpGs) and a list of CVD- and mortality-related traits as outcomes. We utilized our in-house developed analytical pipeline called MR-Seek (https://github.com/OpenOmics/mr-seek.git) to perform the analysis. The full summary statistics of different GWAS datasets were downloaded from NHGRI-EBI. The list of CVD- and mortality- related traits and list of references of those GWAS results are included in Supplementary Data 28. Previously identified *cis*-methylation quantitative trait loci (*cis*-mQTL) were utilized as instrumental variables (IVs) in the MR analysis[20]. The *cis*-mQTLs that were also identified as *trans*-mQTLs were excluded to avoid potential pleiotropic effects. For each CpG site, the IVs comprised independent *cis*-mQTLs pruned for linkage disequilibrium (LD) with an $r^2 < 0.01$. Only one *cis*-mQTL variant with the lowest SNP-CpG *p*-value was retained in each LD block. For CpGs with more than one IV, Inverse-variance weighted (IVW) MR tests were conducted. Heterogeneity and MR-EGGER pleiotropy tests were employed to assess the validity of IVs. Results with a significance level of $P < 0.05$ were excluded. If a CpG has only one instrumental variable (IV), the Wald MR method was applied. Significance levels of MR results were determined based on the Benjamini-Hochberg corrected FDR adjusted *p*-value with a threshold of <0.05.

## Reporting summary
Further information on research design is available in the Nature Portfolio Reporting Summary linked to this article.

## Data availability
The whole genome sequencing (WGS), DNA methylation, RNA sequencing data, and phenotypic data from the Framingham Heart Study (FHS), the Jackson Heart Study (JHS), the Cardiovascular Health Study (CHS), and the Atherosclerosis Risk in Communities (ARIC) study have been deposited in the dbGaP database under accession codes, phs000007 (FHS) [https://www.ncbi.nlm.nih.gov/projects/gap/cgi-bin/study.cgi?study_id=phs000007.v34.p15], phs000964 (JHS) [https://www.ncbi.nlm.nih.gov/projects/gap/cgi-bin/study.cgi?study_id=phs000964.v5.p1], phs001368 (CHS) [https://www.ncbi.nlm.nih.gov/projects/gap/cgi-bin/study.cgi?study_id=phs001368.v4.p2], phs000668 (ARIC) [https://www.ncbi.nlm.nih.gov/projects/gap/cgi-bin/study.cgi?study_id=phs000668.v6.p2]. The WGS, DNA methylation, and phenotypic data from all cohorts used in this study are available under restricted access to protect participant privacy and ensure confidentiality. Access can be obtained by submitting an ancillary study proposal and obtaining IRB approval. Timelines for the approval process range from 4–9 weeks for CHS and 3–6 weeks for ARIC ancillary studies, with specific criteria and proposal forms for the respective studies available at https://chs-nhlbi.org/node/6222 and https://sites.cscc.unc.edu/aric/ancillary-studies-pfg. Full summary statistics for several figures are available in two Zenodo repositories: Fig. 2, Supplementary Figs. 1–3, and Fig. 1 from Responses to Reviewers' Comments File [https://doi.org/10.5281/zenodo.14712757][53] and Supplementary Fig. 4 [https://doi.org/10.5281/zenodo.14712522][54]. Source data are provided with this paper.

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

## Acknowledgements

The views expressed in this manuscript are those of the authors and do not represent the views of the National Heart, Lung, and Blood Institute; the National Institutes of Health; or the U.S. Department of Health and Human Services. The Framingham Heart Study research conducted by S.K., T.H., R.J., N.C., J.M., and D.L. is funded by National Institutes of Health contract N01-HC-25195 and HHSN268201500001I. For this study, the laboratory work was funded by the Division of Intramural Research, National Heart, Lung, and Blood Institute, National Institutes of Health, while the analytical work was funded by the Division of Intramural Research, National Heart, Lung, and Blood Institute, and the Center for Information Technology, National Institutes of Health, Bethesda, MD. The Jackson Heart Study (JHS) research conducted by K.F., L.L., L.R., and A.C. is supported and conducted in collaboration with Jackson State University (HHSN268201800013I), Tougaloo College (HHSN268201800014I), the Mississippi State Department of Health (HHSN268201800015I) and the University of Mississippi Medical Center (HHSN268201800010I, HHSN268201800011I and HHSN268201800012I) contracts from the National Heart, Lung, and Blood Institute (NHLBI) and the National Institute for Minority Health and Health Disparities (NIMHD). The authors also wish to thank the staffs and participants of the JHS. Genome sequencing (dbGap accession phs000964) was performed at the Northwest Genomics Center (HHSN268201100037C). Core support including centralized genomic read mapping and genotype calling, along with variant quality metrics and filtering were provided by the TOPMed Informatics Research Center (R01HL117626; contract HHSN268201800002I). Core support including phenotype harmonization, data management, sample-identity QC, and general program coordination were provided by the TOPMed Data Coordinating Center (R01HL120393; U01HL120393; contract HHSN268201800001I). Cardiovascular Health Study research conducted by study authors J.A.B., B.P., J.F., and M.U. was supported by NHLBI contracts HHSN268201200036C, HHSN268200800007C, HHSN268201800001C, N01HC55222, N01HC85079, N01HC85080, N01HC85081, N01HC85082, N01HC85083, N01HC85086, 75N92021D00006; and NHLBI grants U01HL080295, R01HL087652, R01HL105756, R01HL103612, R01HL120393, and U01HL130114 with additional contribution from the National Institute of Neurological Disorders and Stroke (NINDS). Additional support was provided through R01AG023629 from the National Institute on Aging (NIA). A full list of principal CHS investigators and institutions can be found at CHS-NHLBI.org. Study authors C.B., N.N., J.B., M.F., B.Y. are part of the Atherosclerosis Risk in Communities study. This study has been funded in whole or in part with Federal funds from the National Heart, Lung, and Blood Institute, National Institutes of Health, Department of Health and Human Services, under Contract nos. (75N92022D00001, 75N92022D00002, 75N92022D00003, 75N92022D00004, 75N92022D00005). The authors thank the staff and participants of the ARIC study for their important contributions. J.V.A. is supported by T32 GM080178. P.N. is supported by grants from the R01HL168894 and R01HL148050. The authors thank the Cooperative Centers of Excellence in Hematology (CCEH) supported by NIDDK Grant # DK106829 who helped with procurement of human peripheral blood stem cells. We also thank the flow cytometry core at the United States Department of Veterans Affairs, Tennessee Valley Healthcare System, Nashville, TN.

## Author contributions

S.K., T.H., and D.L. conceived of and designed the study, with input from K.C. and P.N. T.H., S.K., R.J., K.F., L.L., and L.R conducted the analyses in the FHS and JHS cohorts. J.A. and D.O. performed the CHIP model validation experiments. F.P. performed the bioinformatic analyses for the CHIP model validation experiments. J.A.B., J.B, N.S., M.F., J.F., B.P., and C.B. collected and pre-processed previously published data in the CHS or ARIC cohorts that was analyzed in this study. P.N., A.B., and J.W. performed the calling of CHIP in all samples. B.Y. and N.N. conducted analyses in the ARIC cohort. M.U. led a previous study of which the published datasets were used in the current study. S.K and T.H. drafted the manuscript. D.L., L.R., K.F., and A.B. substantively revised the work. A.C. supervised analyses in the JHS cohort. N.C. and J.M. contributed to revisions of the manuscript. These authors contributed equally: S.K., T.H. These authors jointly supervised this work: A.B., D.L. All authors have read and approved the final version of the manuscript.

## Funding

## Competing interests

F.P. is an employee of Biomodal. L.M.R serves as a consultant for the NHLBI TOPMed Administrative Coordinating Center (through Westat). P.N. reports research grants from Allelica, Amgen, Apple, Boston Scientific, Genentech / Roche, and Novartis, personal fees from Allelica, Apple, AstraZeneca, Blackstone Life Sciences, Creative Education Concepts, CRISPR Therapeutics, Eli Lilly & Co, Foresite Labs, Genentech / Roche, GV, HeartFlow, Magnet Biomedicine, Merck, and Novartis, scientific advisory board membership of Esperion Therapeutics, Preciseli, TenSixteen Bio, and Tourmaline Bio, scientific co-founder of TenSixteen Bio, equity in MyOme, Preciseli, and TenSixteen Bio, and spousal employment at Vertex Pharmaceuticals, all unrelated to the present work. B.P. serves on the Steering Committee of the Yale Open Data Access Project funded by Johnson & Johnson. The remaining authors – S.K., T.H., J.A., R.J., M.U., N.N., B.Y., J.A.B., M.F., J.B., N.S., D.O., J.F., C.B., L.R., K.C., J.W., A.C., L.L., K.F., N.H., J.M., A.B., and D.L. – declare no competing interests.

## Additional information

[1]Framingham Heart Study, Framingham, MA 01702, USA. [2]Population Sciences Branch, Division of Intramural Research, National Heart, Lung, and Blood Institute, National Institutes of Health, Bethesda, MD 20892, USA. [3]Division of Genetic Medicine, Department of Medicine, Vanderbilt University Medical Center, Nashville, TN 37232, USA. [4]Medical and Population Genetics and Cardiovascular Disease Initiative, Broad Institute of Harvard and MIT, Cambridge, MA 02142, USA. [5]Cardiovascular Research Center, Massachusetts General Hospital, Boston, MA 02114, USA. [6]Department of Epidemiology, Human Genetics, and Environmental Sciences, School of Public Health, The University of Texas Health Science Center at Houston, Houston, TX 77030, USA. [7]Cardiovascular Health Research Unit, Department of Medicine, University of Washington, Seattle, WA 98101, USA. [8]Brown Foundation Institute of Molecular Medicine, McGovern Medical School, University of Texas Health Science Center at Houston, Houston, TX 77030, USA. [9]Human Genetics Center, School of Public Health, University of Texas Health Science Center at Houston, Houston, TX 77030, USA. [10]Biomodal, The Trinity Building, Chesterford Research Park, Cambridge CB10 1XL, UK. [11]Department of Epidemiology, University of Washington, Seattle, WA 98101, USA. [12]Department of Medicine, Baylor College of Medicine, Houston, TX 77030, USA. [13]Department of Health Systems and Population Health, University of Washington, Seattle, WA 98101, USA. [14]Department of Genetics, University of North Carolina, Chapel Hill, NC 27599, USA. [15]Cardiovascular Research Center and Center for Genomic Medicine, Massachusetts General Hospital, Boston, MA 02114, USA. [16]Department of Medicine, Harvard Medical School, Boston, MA 02115, USA. [17]Department of Human Genetics, Emory University School of Medicine, Atlanta, GA 30322, USA. [18]Department of Medicine, University of Mississippi Medical Center, Jackson, MS 39216, USA. [19]Department of Medicine, University of Colorado at Denver, Aurora, CO 80045, USA. [20]Department of Neurology, Boston University School of Medicine, Boston, MA 02118, USA. [21]Department of Medicine, Section of General Internal Medicine, Boston University School of Medicine and Boston Medical Center, Boston, MA 02118, USA. [22]These authors contributed equally: Sara Kirmani, Tianxiao Huan. [23]These authors jointly supervised this work: Alexander G. Bick, Daniel Levy. ✉e-mail: Tianxiao.huan@nih.gov; Alexander.bick@vumc.org; Levyd@nih.gov

