## [Transparent Peer Review file · Nature Communications]

Epigenome-wide DNA Methylation Association Study of CHIP Provides Insight into Perturbed Gene Regulation

Corresponding Author: Dr Daniel Levy

Version 0:

Reviewer comments:

Reviewer #1

(Remarks to the Author)

Kirmani et al. have performed a meta-EWAS for CHIP in ~8k samples (450k and EPIC). CHIP was identified via Mutect2 in WGS in 3 cohorts (FHS, JHS, and CHS) and via Exome-seq in the ARIC cohort - with the requirement of a variant allele frequency (VAF) 2% in peripheral blood and absence of hematologic malignancy. The EWAS was performed for the 3 major mutational subtypes (DNMT3A, TET2, ASXL1) and identified ~6k associated CpGs in each, with an overlapping total of ~ <10k at $p < 1E-7$. A sensitivity analysis increasing the VAF to 10% recapitulated ~78% of these CpGs.

The majority of the CpG effects were in trans - consistent with the global action of these mutated epigenomic machinery genes. The expected opposite directional effects of TET2 and DNMT3A mutants were observed, with predominately hypermethylation and hypomethylation, respectively.

The downstream effects of these CHIP-associated CpGs were interrogated, including identifying expression quantitative trait methylation (eQTM) and functionally via human hematopoietic stem cell (HSC) CRISPR models. High concordance between the meta-EWAS and human-edited CHIP HSCs was observed. Causal inference via two-sample Mendelian Randomization (MR) identified 261 CpGs associated with CVD traits and all-cause mortality.

In this work, the authors have extended previous analysis in a subset of these data (Uddin et al.) and include additional functional evaluation. I have no major methodological issues with the analysis but highlight some points for the authors to consider below.

Major

1. The authors state in the introduction that CHIP is associated with cardiovascular disease (CVD). They need to include comment/interpretation on the DeCODE findings that the association of smoking with CH itself solely drives this observation [1].
2. The Pathway Enrichment method employed does not appear to have corrected for the known DNAm array bias. A methodology that corrects for this needs to be implemented for accurate enrichment [2, 3].
3. Could the authors comment further on the ~10% of TET mutant CpGs showing decreased methylation – was there any pathway and/or genomic loci enrichment for these findings or potential enrichment for false positives (e.g., how did the p-value distribution compare with the DNMT3A result)?
4. The ASXL1-associated CpG sites showed no significant hits in ASXL1-engineered cells. The authors speculated this may be due to the limited number of ASXL1 CHIP cases in the EWAS. Could the authors comment if there could be any other biological explanation for this?
5. Across all CHIP cases/subtypes expressed genes associated with a CHIP-associated CpG were enriched for immune function. Whilst most of the cohort EWAS included cell-type proportion covariates - could uncorrected subtle effects in cell-type proportion due to clonal selection in these immune cell types be contributing to this enriched immune function?

6. 261 CHIP-associated CpG were identified via 2-step MR to be putatively causally associated with CVD-related traits and/or all-cause mortality. Considering the DeCODE results - could the authors comment on the weakness of the 2-step MR approach, i.e., appropriate statistic thresholds, pleiotropy, etc - as even genetic MR analysis acknowledges the need for further rigour [4].
7. Regarding the FADS2 cg11250194 finding and the comment that hypomethylation of this CpG may be linked to higher triglyceride levels – was this CpG identified previously in any lipid-associated EWAS? [5]
8. Although the EWAS included age as a covariate - a proportion of the common CpG changes identified in all 3 mutants could still be ageing and not CHIP mutant-related. What is the overlap with recent ageing meta-EWASs? [6, 7]
9. As the authors acknowledge, issues around power affect the ability to evaluate less frequent CHIP-associated mutants. Could the authors include comment on potential 'driverless' CH caused by epigenetic changes alone as discussed by Bernstein et al. [8]
10. In the Discussion the authors speculate that an unhealthy diet may be associated with CHIP through epigenetic mechanisms. Could this be confounded by concurrent poor diet with smoking?

1. Stacey, S.N., et al., Genetics and epidemiology of mutational barcode-defined clonal hematopoiesis. *Nature Genetics*, 2023. 55(12): p. 2149-2159.
2. Phipson, B., J. Maksimovic, and A. Oshlack, missMethyl: an R package for analyzing data from Illumina's HumanMethylation450 platform. *Bioinformatics*, 2016. 32(2): p. 286-8.
3. Maksimovic, J., A. Oshlack, and B. Phipson, Gene set enrichment analysis for genome-wide DNA methylation data. *Genome Biol*, 2021. 22(1): p. 173.
4. Reed, Z.E., et al., Exploring pleiotropy in Mendelian randomisation analyses: What are genetic variants associated with 'cigarette smoking initiation' really capturing? *Genet Epidemiol*, 2024.
5. Dekkers, K.F., et al., Blood lipids influence DNA methylation in circulating cells. *Genome Biol*, 2016. 17(1): p. 138.
6. Bernabeu, E., et al., Refining epigenetic prediction of chronological and biological age. *Genome Medicine*, 2023. 15(1): p. 12.
7. Seale, K., et al., A comprehensive map of the ageing blood methylome. *bioRxiv*, 2023: p. 2023.12.20.572666.
8. Bernstein, N., et al., Analysis of somatic mutations in whole blood from 200,618 individuals identifies pervasive positive selection and novel drivers of clonal hematopoiesis. *Nat Genet*, 2024. 56(6): p. 1147-1155.

Reviewer #2

(Remarks to the Author)

The authors have presented a thorough and well-written paper exploring the role of methylation in a variety of CHIP subtypes. They linked CHIP to alterations in specific CpGs through an epigenome-wide association study (EWAS) and then validated these changes using an in vitro model - a step that added considerable depth to their analysis. The authors then linked CHIP-associated CpGs with gene expression to explore the functional role of these methylation changes. Finally, they performed a Mendelian Randomisation Analysis to identify putative causal CpGs linked to cardiovascular outcomes. Overall, we found the methodology generally sound and our concerns are relatively minor. That said, we have a few comments which are outlined below:

- The EWAS looks fine to us, but we're not experts in cross-ancestry analyses. An additional reviewer might be considered who is an expert in such fields.
- In the in vitro validation analysis, the authors focussed only on the subset of CpG sites that were significantly associated with CHIP in both the EWAS analysis and nominally differentially methylated in the in vitro model of CHIP. 1) Why was this approach taken? In other words, why is it appropriate to ignore the CpGs that appear in only one set? 2) Similarly, while the directionality overlap in the subset of CpGs that change in both analyses looks generally quite good, why do we see such a small percentage of the EWAS hits captured in the experimental data in the first place?
- In the in vitro validation analysis, the -ASXL1-associated CpG sites showed no significant hits in ASXL1-engineered cells. We would like to see this validation failure discussed more clearly in the limitations of downstream analyses with this type of CHIP, i.e. does this affect the interpretation of any results?
- In the "Association of DNA Methylation with Gene Expression and Pathway Analyses" section, it might be interesting to see how the directionality of gene expression changes aligns with methylation changes.
- Additionally, also in the "Association of DNA Methylation with Gene Expression and Pathway Analyses" section, gene expression changes within 1Mb of the CpG in question were analysed. This is quite a large distance – the authors may also want to consider smaller distances, e.g. 100kb, which may help increase the specificity of their results.
- The section "Association of DNA Methylation with Genetic Variants and Mendelian Randomization Analysis" needs some more detail (perhaps just in Methods) regarding the first two paragraphs. E.g. specifically, how were mQTLs linked with diseases?
- We would also like to see some more discussion of the potential limitations of the Mendelian Randomisation approach in identifying causal CpGs. In particular, do the authors believe that cis-mQTLs satisfy the "exclusion restriction" assumption of an instrumental variable? In other words, is it possible/likely that the cis-mQTLs might be affecting the outcome through pathways other than via the exposure? If so, how does this affect the interpretation of the results?
- The legend of Fig 3 doesn't seem quite right – it states "Dot plots of methylation status from -1.0 (no methylation) to 1.0 (complete methylation)", but it seems it should probably refer to methylation change.

Reviewer #3

(Remarks to the Author)

Reviewer #4

(Remarks to the Author)

RE: Kirmani et al.

Epigenome-wide DNA Methylation Association Study of CHIP Provides Insight into Perturbed Gene Regulation

Summary

Kirmani et al. present an epigenetic characterization of CHIP compiled from several large epidemiological sequencing cohorts. By looking at DNA methylation profiles from individuals with CHIP, the authors try to deduce some profiles that may link the mutations to regulation of epigenetic changes that drive some CHIP-related pathologies. There is also attempt to connect differentially methylated CpGs to downstream gene expression changes, although nothing is really significant. The manuscript is essentially descriptive and several limitations impact the relevance of the study. It is not clear what the real conclusion from this analysis is or how it contributes to our understanding of how the mutant cells gain clonal dominance (which as outlined by the authors in the introduction is part of the motivation for the study). In the current state, it is uncertain what information can be obtained from this that advances the field of CHIP.

Major Points:

- The genomic, DNA methylation and gene expression profiles are obtained from whole blood samples. The fraction of mutated cells in this sample will be highly variable and typically very low. While it is certainly impressive that certain DNA methylation patterns can be associated with DNMT3A, TET2, and ASXL1 mutations given this, how do the authors account for VAF of mutation with altered DNA methylation?
- Given it is known that these common CHIP mutations alter the differentiation spectrum of mutant stem and progenitor cells, the cellular composition of each individual with different mutations is going to be highly variable. How can the authors account for hematopoietic cell type specific DNA methylation differences? DNA methylation is essentially a measure of cell type identity. And the DNA methylation profile of a neutrophil is very different from a T-cell. Any changes in DNA methylation may simply reflect different proportions of mature hematopoietic cell types that went into the sequencing pool.
- The authors engineer DNMT3A, TET2, and ASXL1 mutations into healthy CD34+ cells and then perform DNA methylation profiling. They mostly recover the DNA methylation profiles from the sequenced CHIP donors. It is unclear what additional value is gained from this experiment. How does this inform of the mechanism by which these mutations change DNA methylation profile and skew cell fate decisions?
- There is a largely no correlation between altered DNA methylation patterns and changes in gene expression profile (line 211). Is there any information that can be obtained to inform how altered epigenomic profile of CHIP cells alters their function?
- Line 77-79: "The novel genes and pathways linked to the epigenetic features of CHIP may serve as therapeutic targets for preventing or treating CHIP-mediated diseases." This is very speculative. Please be more specific about what novel genes / pathways have been identified by this study for this purpose. And how might they serve as therapeutic targets in the future?

Version 1:

Reviewer comments:

Reviewer #1

(Remarks to the Author)

>>> Reviewer #1 Replies

Kirmani et al. have performed a meta-EWAS for CHIP in ~8k samples (450k and EPIC). CHIP was identified via Mutect2 in WGS in 3 cohorts (FHS, JHS, and CHS) and via Exome-seq in the ARIC cohort - with the requirement of a variant allele frequency (VAF) 2% in peripheral blood and absence of hematologic malignancy. The EWAS was performed for the 3 major mutational subtypes (DNMT3A, TET2, ASXL1) and identified ~6k associated CpGs in each, with an overlapping total of ~<10k at $p < 1E-7$. A sensitivity analysis increasing the VAF to 10% recapitulated ~78% of these CpGs.

The majority of the CpG effects were in trans - consistent with the global action of these mutated epigenomic machinery genes. The expected opposite directional effects of TET2 and DNMT3A mutants were observed, with predominately hypermethylation and hypomethylation, respectively.

The downstream effects of these CHIP-associated CpGs were interrogated, including identifying expression quantitative trait methylation (eQTM) and functionally via human hematopoietic stem cell (HSC) CRISPR models. High concordance between the meta-EWAS and human-edited CHIP HSCs was observed. Causal inference via two-sample Mendelian Randomization (MR) identified 261 CpGs associated with CVD traits and all-cause mortality.

In this work, the authors have extended previous analysis in a subset of these data (Uddin et al.) and include additional functional evaluation. I have no major methodological issues with the analysis but highlight some points for the authors to consider below.

Major

1. The authors state in the introduction that CHIP is associated with cardiovascular disease (CVD). They need to include comment/interpretation on the DeCODE findings that the association of smoking with CH itself solely drives this observation [1].

Response: We thank the reviewer for this suggestion. We have revised the manuscript to include discussion about the DeCODE study and their finding that smoking drove the association between CHIP and CVD. The revised manuscript is changed as follows (see lines 348-370, page 10):

“Although smoking status was included as a covariate in the statistical models for all study cohorts, there could be residual confounding as smoking behavior may not be fully adjusted for in the analysis. Thus, smoking could still be driving the association between CHIP and CVD, as was reported in a recent study of CH.³² Of note, there are several studies across diverse populations and in different settings that controlled for smoking as a covariate and also found an association of CHIP (a specific form of CH) with CVD. For instance, in a recent study, Diez-Diez et al.³³ clarified the directionality of the CHIP-CVD relationship with adjustment for smoking and concluded that CH confers an increased risk of developing atherosclerosis.

Additionally, the way that the DeCODE³² investigators ascertained “CH” may have contributed to their finding that CH was not associated with CVD, as it is distinct from the definition of “CHIP.” CH includes clonal events with known leukemic driver gene mutations, such as CHIP and mosaic chromosomal alterations (mCAs), and clonal events without clear driver genes. Given recent findings that each of these distinct classes of CH has unique phenotypic consequences,^{3,34} the lack of association between CH and CHIP reported by Stacey et al. may be due to the grouping of heterogeneous CH subtypes. Moreover, it has been previously demonstrated that small changes in the stringency with which CHIP is ascertained can have an outsized effect on downstream analyses. For example, Vlasschaert et al.³⁵ reported the importance of CHIP detection stringency in relation to CHIP-associated CVD risk. Specifically, more stringent criteria (≥ 5 supporting reads) were associated with CVD risk, while less stringent criteria (≥ 3 supporting reads) attenuated the association of CHIP with CVD. Their study provides an up-to-date and nuanced explanation of the CHIP-CVD relationship.”

>>> Great answer to this – thanks.

2. The Pathway Enrichment method employed does not appear to have corrected for the known DNAm array bias. A methodology that corrects for this needs to be implemented for accurate enrichment [2, 3].

Response: Thank you for the suggestion. In the revised manuscript, we have now corrected the GO enrichment analysis for known DNAm array bias, ⁵⁷ as recommended. Specifically, we used the missMethyl R package to adjust for this bias, and have updated the manuscript to replace the previous GO enrichment results with the revised analysis. Results have changed as follows: (see lines 193-204, page 6):

“For any CHIP, DNMT3A CHIP, and ASXL1 CHIP the enriched GO terms related to broad cellular developmental and organismal processes, while for TET2 CHIP the top GO terms related to cellular regulation and cell signaling (Supplementary Tables 14-17). For example, for any CHIP and the two most common CHIP subtypes, the top ten most significant ontologies included multicellular organism development, anatomical structure development, system development, and developmental process. For TET2 CHIP, the most significant ontology terms related to a cellular response to stimulus, regulation of cellular processes, and cell signaling. Notably, for the genes annotated to the 554 TET2-associated CpGs that were found to be demethylated, the top GO terms were enriched for cellular developmental and organismal processes such as multicellular organism development and system development – similar to the enriched GO terms for genes annotated to DNMT3A CHIP-associated CpGs (Supplementary Table 29).”

>>> It is good that the authors have performed this non-biased reanalysis

3. Could the authors comment further on the ~10% of TET mutant CpGs showing decreased methylation – was there any

pathway and/or genomic loci enrichment for these findings or potential enrichment for false positives (e.g., how did the p-value distribution compare with the DNMT3A result)?

Response: Thank you for your insightful suggestion. We performed GO and pathway enrichment analysis on the 10% of TET2 mutant CpGs showing decreased methylation. The findings have been incorporated into the manuscript (see lines 200-204, page 6):

“Notably, for the genes annotated to the 554 TET2-associated CpGs that were found to be demethylated, the top GO terms were enriched for cellular developmental and organismal processes such as multicellular organism development and system development – similar to the enriched GO terms for genes annotated to DNMT3A CHIP-associated CpGs (Supplementary Table 29).”

To address the possibility of false-positive enrichment, we compared the p-value distributions of the TET2-associated CpGs with decreased methylation to those of DNMT3A-associated CpGs (see Response Letter Figure 1). This comparison revealed no discernible patterns in the p-value distributions, indicating distinct effects of TET2 CHIP and DNMT3A CHIP on overlapping CpGs. These findings suggest that the demethylated CpGs associated with TET2 mutations are unlikely to be enriched for false positives. Instead, the weak or inconsistent correlation observed between TET2- and DNMT3A-associated demethylated CpGs further supports the independent roles of these mutations in methylation regulation.

Figure 1. Comparison of P-value Distributions Between Demethylated TET2 CpGs and DNMT3A CpGs

>>> Ok

4. The ASXL1-associated CpG sites showed no significant hits in ASXL1-engineered cells. The authors speculated this may be due to the limited number of ASXL1 CHIP cases in the EWAS. Could the authors comment if there could be any other biological explanation for this?

Response: The revised manuscript includes comments on potential biological explanations for ASXL1-associated CpGs showing no significant enrichment in ASXL1-engineered cells in the Discussion section (see lines 277-288, page 8):

“The lack of enrichment of ASXL1-associated CpGs in the ASXL1-engineered cells may limit the validity of the study’s downstream analyses with ASXL1 CHIP. This observation may be due to the limited number of ASXL1 CHIP cases in the EWAS as well as several biological factors. ASXL1 mutations primarily affect histone modifications, particularly H2AK119 ubiquitination, which indirectly influences chromatin accessibility.²⁶ Recent studies have shown that ASXL1 loss-of-function mutations increase chromatin accessibility, potentially resulting in individualistic methylation changes influenced by other genetic and environmental factors.²⁷ Furthermore, the effects of ASXL1 mutations on methylation might be temporally dynamic or cell-type specific, aspects not fully captured in our current experimental design. Future studies with larger sample sizes and longer observation periods may help elucidate the complex relationship between ASXL1 mutations and DNA methylation patterns in CHIP.”

>>> Ok

5. Across all CHIP cases/subtypes expressed genes associated with a CHIP-associated CpG were enriched for immune function. Whilst most of the cohort EWAS included cell-type proportion covariates - could uncorrected subtle effects in cell-type proportion due to clonal selection in these immune cell types be contributing to this enriched immune function?

Response: Thank you for raising this important point. Although we included cell-type proportions as covariates in this study, we cannot exclude the possibility that subtle uncorrected effects in cell-type composition due to clonal selection in immune cells could potentially contribute to this observed enrichment. In the revised manuscript, we acknowledge that while cell-type adjustments reduce these confounding effects, residual contributions from altered immune cell proportions remain possible. See the revised Discussion (lines 399-405, page 11):

“Last, although cell-type proportions were included as covariates for all cohorts, we cannot exclude the possibility that subtle uncorrected effects in cell-type proportions due to clonal selection in immune cells may contribute to the enrichment of immune function observed for TET2 and ASXL1 CHIP eGenes. While cell-type adjustments reduce confounding effects, residual contributions from altered immune cell proportions remain possible. Future studies investigating cell-type specific DNA methylation and gene expression may provide additional clarity on the impact of CHIP on immune gene expression”.

>>> Ok great

6. 261 CHIP-associated CpG were identified via 2-step MR to be putatively causally associated with CVD-related traits and/or all-cause mortality. Considering the DeCODE results - could the authors comment on the weakness of the 2-step MR

approach, i.e., appropriate statistic thresholds, pleiotropy, etc - as even genetic MR analysis acknowledges the need for further rigour [4].

Response: We thank the reviewer for raising this important consideration. In our analysis, we used the inverse variance weighted (IVW) method for MR to explore potential causal relationships between CHIP-associated CpGs and CVD traits or all-cause mortality, using independent cis-mQTLs as instrumental variables. To minimize horizontal pleiotropy, we initially excluded cis-mQTLs that also acted as trans-mQTLs. For CpGs with three or more independent IVs (defined as cis-mQTLs with LD < 0.01 within a 2MB cis-region of the CpG), we further assessed pleiotropy through MR-Egger, where a non-zero intercept would suggest potential pleiotropy, and excluded results with P values < 0.05. Despite these precautions, the two-sample MR approach has known limitations and current methods for detecting and addressing pleiotropy may be ineffective under some plausible scenarios. As noted in reference 4 from Reviewer #1, genetic instruments, especially for complex traits such as smoking, can show horizontal pleiotropy for multiple traits that are unlikely to be causally linked to the exposure, suggesting possible pleiotropic influences. We acknowledge that longitudinal and functional studies are critical for reinforcing causal findings and we now discuss these limitations in the revised manuscript (see lines 385-390, page 11):

“Despite our attempt to minimize horizontal pleiotropy by excluding cis-mQTLs that also serve as trans-mQTLs and excluding CpGs with three or more independent instrumental variables through MR-Egger using a threshold of P values <0.05, the two-sample MR approach has known limitations.³⁷ Methods for detecting and addressing pleiotropy may be ineffective³⁸ and, thus, longitudinal and functional studies are needed to reinforce causal findings.”

>>> Ok great

7. Regarding the FADS2 cg11250194 finding and the comment that hypomethylation of this CpG may be linked to higher triglyceride levels – was this CpG identified previously in any lipid-associated EWAS? [5]

Response: Yes, CpG site - cg11250194 – was identified in an epigenome-wide association study of lipid-related metabolic measures performed by Gomez-Alonso et al³⁰ (see lines 317-318, page 9):

“Moreover, cg11250194 was previously identified in an EWAS of lipid-related metabolic measures.³⁰”

>>> Great good to include this info

8. Although the EWAS included age as a covariate - a proportion of the common CpG changes identified in all 3 mutants could still be ageing and not CHIP mutant-related. What is the overlap with recent ageing meta-EWASs? [6, 7]

Response: There are 19 CpGs that are common across all three CHIP driver gene mutations – DNMT3A, TET2, and ASXL1 CHIP (Bonferroni-corrected $P < 1 \times 10^{-7}$) from our meta-EWAS of CHIP. When comparing these CpG sites with the CpGs from a recent EWAS of chronological age,¹ we found that none of the CpGs overlapped. We have included this new information in our revised Results section on lines 161-166, page 5:

“Although age was included as a covariate, there remains a possibility that common CpG sites across all three CHIP driver genes may be related to age rather than CHIP mutation. To assess this, we compared the 19 CpGs that are common among all three CHIP driver gene mutations from our meta-EWAS of CHIP with the CpGs from a recent EWAS of chronological age in the Generation Scotland cohort¹⁷ (N=18,413). No CpGs overlapped, suggesting that common CpGs across all CHIP driver genes are related to CHIP mutation rather than age.”

>>> Ok good to exclude this specifically

9. As the authors acknowledge, issues around power affect the ability to evaluate less frequent CHIP-associated mutants. Could the authors include comment on potential ‘driverless’ CH caused by epigenetic changes alone as discussed by Bernstein et al. [8]

Response: We thank the reviewer for this comment. We have included in the revised Discussion section (lines 371-382, pages 10-11) comments on the potential ‘driverless’ CH caused by epigenetic changes as follows:

“Driverless CH is the occurrence of clonal expansions in blood without a known CHIP driver mutation and is estimated to drive the majority of clonal expansions in the elderly.³⁶ Bernstein et al.³⁶ identified regions within exome sequences that are under positive selection to identify additional driver mutations in whole blood (large clones >0.1) together with validation of positive selection in single cell-derived hematopoietic myeloid and lymphoid colonies. The inclusion of mutations in these fitness-inferred CH genes increases prevalence of CH by 18% in the UKBB cohort. In our study, CHIP was defined when an individual harbored at least one deleterious insertion/deletion or single nucleotide variant in any of the 74 genes that have been previously linked to myeloid malignancy at a variant allele frequency of at least 2%. Given the study’s scope, we did not include driverless CH in our CHIP definition. Thus, CHIP prevalence in our study

may be underestimated relative to studies that account for driverless clonal expansion.”

>>> Ok, good to include

10. In the Discussion the authors speculate that an unhealthy diet may be associated with CHIP through epigenetic mechanisms. Could this be confounded by concurrent poor diet with smoking?

Response: We thank the reviewer for raising this point. We included smoking status as a covariate in the models for all participating cohorts, which reduces the likelihood that smoking alone accounts for the observed association between CHIP and CpG sites. Ma et al.²⁹ also used models that adjusted for smoking status when identifying diet-associated CpG sites. However, we recognize that our cross-sectional study design may limit the ability to fully separate the effects of diet from smoking on CHIP risk, as cross-sectional analyses capture associations at a single time point without assessing how these exposures and their interactions may change over time. Consequently, some of the observed association between unhealthy diet and CHIP through epigenetic mechanisms may still reflect residual smoking effects. We have added this consideration to the revised Discussion section (see lines 325-330, page 9):

“Despite including smoking status as a covariate in the models for all participating cohorts in both studies,²⁹ we recognize that our ability to fully separate the effects of diet from smoking on CHIP risk may be limited due to our cross-sectional study design. As a result, some of the observed association between unhealthy diet and CHIP through epigenetic mechanisms may reflect residual smoking effects.”

>>> Ok great to qualify this

Reviewer #2

(Remarks to the Author)

The authors have addressed our concerns.

Reviewer #3

(Remarks to the Author)

Reviewer #4

(Remarks to the Author)

The authors have done a thorough job at responding to previous reviewer comments. Technically they have answered all the concerns raised in the prior review.

However, there is a difference between statistical significance and biological significance. The power in this study comes from leveraging large patient groups to achieve high sample numbers required for robust statistical comparisons. However, the authors have still not clearly accounted for biological diversity. Although the statistical justification of cell type correction is robust and justified, how this plays out in a real world setting is not validated. Similarly, it is still dubious how evaluation of individual CpGs is meaningful given the complex nature of DNA methylation. Differentially methylated regions would be more meaningful. So while this is an excellent study leveraging a large patient populations, the real biological implications of this analysis remain unclear.

Responses to Reviewers' Comments

The reviewer comments are in *italics* and responses are in plain font.

Reviewer #1 (Remarks to the Author):

Kirmani et al. have performed a meta-EWAS for CHIP in ~8k samples (450k and EPIC). CHIP was identified via Mutect2 in WGS in 3 cohorts (FHS, JHS, and CHS) and via Exome-seq in the ARIC cohort - with the requirement of a variant allele frequency (VAF) $\geq 2\%$ in peripheral blood and absence of hematologic malignancy. The EWAS was performed for the 3 major mutational subtypes (DNMT3A, TET2, ASXL1) and identified ~6k associated CpGs in each, with an overlapping total of $\sim < 10k$ at $p < 1E-7$. A sensitivity analysis increasing the VAF to $\geq 10\%$ recapitulated ~78% of these CpGs.

The majority of the CpG effects were in trans - consistent with the global action of these mutated epigenomic machinery genes. The expected opposite directional effects of TET2 and DNMT3A mutants were observed, with predominately hypermethylation and hypomethylation, respectively. The downstream effects of these CHIP-associated CpGs were interrogated, including identifying expression quantitative trait methylation (eQTM) and functionally via human hematopoietic stem cell (HSC) CRISPR models. High concordance between the meta-EWAS and human-edited CHIP HSCs was observed. Causal inference via two-sample Mendelian Randomization (MR) identified 261 CpGs associated with CVD traits and all-cause mortality.

In this work, the authors have extended previous analysis in a subset of these data (Uddin et al.) and include additional functional evaluation. I have no major methodological issues with the analysis but highlight some points for the authors to consider below.

Major

1. The authors state in the introduction that CHIP is associated with cardiovascular disease (CVD). They need to include comment/interpretation on the DeCODE findings that the association of smoking with CH itself solely drives this observation [1].

Response: We thank the reviewer for this suggestion. We have revised the manuscript to include discussion about the DeCODE study and their finding that smoking drove the association between CHIP and CVD.³² The revised manuscript is changed as follows (see lines 348-370, page 10):

“Although smoking status was included as a covariate in the statistical models for all study cohorts, there could be residual confounding as smoking behavior may not be fully adjusted for in the analysis. Thus, smoking could still be driving the association between CHIP and CVD, as was reported in a recent study of CH.³² Of note, there are several studies across diverse populations and in different settings that controlled for smoking as a covariate and also found an association of CHIP (a specific form of CH) with CVD. For instance, in a recent study, Diez-Diez et al.³³ clarified the directionality of the CHIP-CVD

relationship with adjustment for smoking and concluded that CH confers an increased risk of developing atherosclerosis.

Additionally, the way that the DeCODE³² investigators ascertained “CH” may have contributed to their finding that CH was not associated with CVD, as it is distinct from the definition of “CHIP.” CH includes clonal events with known leukemic driver gene mutations, such as CHIP and mosaic chromosomal alterations (mCAs), and clonal events without clear driver genes. Given recent findings that each of these distinct classes of CH has unique phenotypic consequences,^{3,34} the lack of association between CH and CHIP reported by Stacey et al. may be due to the grouping of heterogeneous CH subtypes.

Moreover, it has been previously demonstrated that small changes in the stringency with which CHIP is ascertained can have an outsize effect on downstream analyses. For example, Vlasschaert et al.³⁵ reported the importance of CHIP detection stringency in relation to CHIP-associated CVD risk. Specifically, more stringent criteria (≥ 5 supporting reads) were associated with CVD risk, while less stringent criteria (≥ 3 supporting reads) attenuated the association of CHIP with CVD. Their study provides an up-to-date and nuanced explanation of the CHIP-CVD relationship.”

32. Stacey SN, Zink F, Halldorsson GH, et al. Genetics and epidemiology of mutational barcode-defined clonal hematopoiesis. *Nat Genet.* 2023;55(12):2149-2159. doi:10.1038/s41588-023-01555-z

2. *The Pathway Enrichment method employed does not appear to have corrected for the known DNAm array bias. A methodology that corrects for this needs to be implemented for accurate enrichment [2, 3].*

Response: Thank you for the suggestion. In the revised manuscript, we have now corrected the GO enrichment analysis for known DNAm array bias,⁵⁷ as recommended. Specifically, we used the *missMethyl* R package to adjust for this bias, and have updated the manuscript to replace the previous GO enrichment results with the revised analysis. Results have changed as follows: (see lines 193-204, page 6):

“For any CHIP, *DNMT3A* CHIP, and *ASXL1* CHIP the enriched GO terms related to broad cellular developmental and organismal processes, while for *TET2* CHIP the top GO terms related to cellular regulation and cell signaling (Supplementary Tables 14-17). For example, for any CHIP and the two most common CHIP subtypes, the top ten most significant ontologies included multicellular organism development, anatomical structure development, system development, and developmental process. For *TET2* CHIP, the most significant ontology terms related to a cellular response to stimulus, regulation of cellular processes, and cell signaling. Notably, for the genes annotated to the 554 *TET2*-associated CpGs that were found to be demethylated, the top GO terms were enriched for cellular developmental and organismal processes such as multicellular organism

development and system development – similar to the enriched GO terms for genes annotated to *DNMT3A* CHIP-associated CpGs (Supplementary Table 29).”].

57. Phipson B, Maksimovic J, Oshlack A. missMethyl: an R package for analyzing data from Illumina's HumanMethylation450 platform. *Bioinformatics*. 2016;32(2):286-288. doi:10.1093/bioinformatics/btv560

3. Could the authors comment further on the ~10% of *TET* mutant CpGs showing decreased methylation – was there any pathway and/or genomic loci enrichment for these findings or potential enrichment for false positives (e.g., how did the p-value distribution compare with the *DNMT3A* result)?

Response: Thank you for your insightful suggestion. We performed GO and pathway enrichment analysis on the 10% of *TET2* mutant CpGs showing decreased methylation. The findings have been incorporated into the manuscript (see lines 200-204, page 6):

“Notably, for the genes annotated to the 554 *TET2*-associated CpGs that were found to be demethylated, the top GO terms were enriched for cellular developmental and organismal processes such as multicellular organism development and system development – similar to the enriched GO terms for genes annotated to *DNMT3A* CHIP-associated CpGs (Supplementary Table 29).”

To address the possibility of false-positive enrichment, we compared the p-value distributions of the *TET2*-associated CpGs with decreased methylation to those of *DNMT3A*-associated CpGs (see Response Letter Figure 1). This comparison revealed no discernible patterns in the p-value distributions, indicating distinct effects of *TET2* CHIP and *DNMT3A* CHIP on overlapping CpGs. These findings suggest that the demethylated CpGs associated with *TET2* mutations are unlikely to be enriched for false positives. Instead, the weak or inconsistent correlation observed between *TET2*- and *DNMT3A*-associated demethylated CpGs further supports the independent roles of these mutations in methylation regulation.

Figure 1. Comparison of P-value Distributions Between Demethylated *TET2* CpGs and *DNMT3A* CpGs

4. *The ASXL1-associated CpG sites showed no significant hits in ASXL1-engineered cells. The authors speculated this may be due to the limited number of ASXL1 CHIP cases in the EWAS. Could the authors comment if there could be any other biological explanation for this?*

Response: The revised manuscript includes comments on potential biological explanations for *ASXL1*-associated CpGs showing no significant enrichment in *ASXL1*-engineered cells in the Discussion section (see lines 277-288, page 8):

“The lack of enrichment of *ASXL1*-associated CpGs in the *ASXL1*-engineered cells may limit the validity of the study’s downstream analyses with *ASXL1* CHIP. This observation may be due to the limited number of *ASXL1* CHIP cases in the EWAS as well as several biological factors. *ASXL1* mutations primarily affect histone modifications, particularly H2AK119 ubiquitination, which indirectly influences chromatin accessibility.²⁶ Recent studies have shown that *ASXL1* loss-of-function mutations increase chromatin accessibility, potentially resulting in individualistic methylation changes influenced by other genetic and environmental factors.²⁷ Furthermore, the effects of *ASXL1* mutations on methylation might be temporally dynamic or cell-type specific, aspects not fully captured in our current experimental design. Future studies with larger sample sizes and longer observation periods may help elucidate the complex relationship between *ASXL1* mutations and DNA methylation patterns in CHIP.”

5. *Across all CHIP cases/subtypes expressed genes associated with a CHIP-associated CpG were enriched for immune function. Whilst most of the cohort EWAS included cell-type proportion covariates - could uncorrected subtle effects in cell-type proportion due to clonal selection in these immune cell types be contributing to this enriched immune function?*

Response: Thank you for raising this important point. Although we included cell-type proportions as covariates in this study, we cannot exclude the possibility that subtle uncorrected effects in cell-type composition due to clonal selection in immune cells could potentially contribute to this observed enrichment. In the revised manuscript, we acknowledge that while cell-type adjustments reduce these confounding effects, residual contributions from altered immune cell proportions remain possible. See the revised Discussion (lines 399-405, page 11):

“Last, although cell-type proportions were included as covariates for all cohorts, we cannot exclude the possibility that subtle uncorrected effects in cell-type proportions due to clonal selection in immune cells may contribute to the enrichment of immune function observed for *TET2* and *ASXL1* CHIP eGenes. While cell-type adjustments reduce confounding effects, residual contributions from altered immune cell proportions remain possible. Future studies investigating cell-type specific DNA methylation and gene expression may provide additional clarity on the impact of CHIP on immune gene expression”.

6. 261 CHIP-associated CpG were identified via 2-step MR to be putatively causally associated with CVD-related traits and/or all-cause mortality. Considering the DeCODE results - could the authors comment on the weakness of the 2-step MR approach, i.e., appropriate statistic thresholds, pleiotropy, etc - as even genetic MR analysis acknowledges the need for further rigour [4].

Response: We thank the reviewer for raising this important consideration. In our analysis, we used the inverse variance weighted (IVW) method for MR to explore potential causal relationships between CHIP-associated CpGs and CVD traits or all-cause mortality, using independent *cis*-mQTLs as instrumental variables. To minimize horizontal pleiotropy, we initially excluded *cis*-mQTLs that also acted as *trans*-mQTLs. For CpGs with three or more independent IVs (defined as *cis*-mQTLs with LD < 0.01 within a 2MB *cis*-region of the CpG), we further assessed pleiotropy through MR-Egger, where a non-zero intercept would suggest potential pleiotropy, and excluded results with P values < 0.05. Despite these precautions, the two-sample MR approach has known limitations and current methods for detecting and addressing pleiotropy may be ineffective under some plausible scenarios. As noted in reference 4 from Reviewer #1, genetic instruments, especially for complex traits such as smoking, can show horizontal pleiotropy for multiple traits that are unlikely to be causally linked to the exposure, suggesting possible pleiotropic influences. We acknowledge that longitudinal and functional studies are critical for reinforcing causal findings and we now discuss these limitations in the revised manuscript (see lines 385-390, page 11):

“Despite our attempt to minimize horizontal pleiotropy by excluding *cis*-mQTLs that also serve as *trans*-mQTLs and excluding CpGs with three or more independent instrumental variables through MR-Egger using a threshold of P values <0.05, the two-sample MR approach has known limitations.³⁷ Methods for detecting and addressing pleiotropy may be ineffective³⁸ and, thus, longitudinal and functional studies are needed to reinforce causal findings.”

7. Regarding the *FADS2* cg11250194 finding and the comment that hypomethylation of this CpG may be linked to higher triglyceride levels – was this CpG identified previously in any lipid-associated EWAS? [5]

Response: Yes, CpG site - cg11250194 – was identified in an epigenome-wide association study of lipid-related metabolic measures performed by Gomez-Alonso et al³⁰ (see lines 317-318, page 9):

“Moreover, cg11250194 was previously identified in an EWAS of lipid-related metabolic measures.³⁰”

30. Gomez-Alonso MDC, Kretschmer A, Wilson R, et al. DNA methylation and lipid metabolism: an EWAS of 226 metabolic measures. *Clin Epigenetics*. 2021;13(1):7. Published 2021 Jan 7. doi:10.1186/s13148-020-00957-8

8. *Although the EWAS included age as a covariate - a proportion of the common CpG changes identified in all 3 mutants could still be ageing and not CHIP mutant-related. What is the overlap with recent ageing meta-EWASs? [6, 7]*

Response: There are 19 CpGs that are common across all three CHIP driver gene mutations – *DNMT3A*, *TET2*, and *ASXL1* CHIP (Bonferroni-corrected $P < 1 \times 10^{-7}$) from our meta-EWAS of CHIP. When comparing these CpG sites with the CpGs from a recent EWAS of chronological age,¹ we found that none of the CpGs overlapped. We have included this new information in our revised Results section on lines 161-166, page 5:

“Although age was included as a covariate, there remains a possibility that common CpG sites across all three CHIP driver genes may be related to age rather than CHIP mutation. To assess this, we compared the 19 CpGs that are common among all three CHIP driver gene mutations from our meta-EWAS of CHIP with the CpGs from a recent EWAS of chronological age in the Generation Scotland cohort¹⁷ (N=18,413). No CpGs overlapped, suggesting that common CpGs across all CHIP driver genes are related to CHIP mutation rather than age.”

9. *As the authors acknowledge, issues around power affect the ability to evaluate less frequent CHIP-associated mutants. Could the authors include comment on potential ‘driverless’ CH caused by epigenetic changes alone as discussed by Bernstein et al. [8]*

Response: We thank the reviewer for this comment. We have included in the revised Discussion section (lines 371-382, pages 10-11) comments on the potential ‘driverless’ CH caused by epigenetic changes as follows:

“Driverless CH is the occurrence of clonal expansions in blood without a known CHIP driver mutation and is estimated to drive the majority of clonal expansions in the elderly.³⁶ Bernstein et al.³⁶ identified regions within exome sequences that are under positive selection to identify additional driver mutations in whole blood (large clones >0.1) together with validation of positive selection in single cell-derived hematopoietic myeloid and lymphoid colonies. The inclusion of mutations in these fitness-inferred CH genes increases prevalence of CH by 18% in the UKBB cohort. In our study, CHIP was defined when an individual harbored at least one deleterious insertion/deletion or single nucleotide variant in any of the 74 genes that have been previously linked to myeloid malignancy at a variant allele frequency of at least 2%. Given the study's scope, we did not include driverless CH in our CHIP definition. Thus, CHIP prevalence in our study may be underestimated relative to studies that account for driverless clonal expansion.”

10. In the Discussion the authors speculate that an unhealthy diet may be associated with CHIP through epigenetic mechanisms. Could this be confounded by concurrent poor diet with smoking?

Response: We thank the reviewer for raising this point. We included smoking status as a covariate in the models for all participating cohorts, which reduces the likelihood that smoking alone accounts for the observed association between CHIP and CpG sites. Ma et al.²⁹ also used models that adjusted for smoking status when identifying diet-associated CpG sites. However, we recognize that our cross-sectional study design may limit the ability to fully separate the effects of diet from smoking on CHIP risk, as cross-sectional analyses capture associations at a single time point without assessing how these exposures and their interactions may change over time. Consequently, some of the observed association between unhealthy diet and CHIP through epigenetic mechanisms may still reflect residual smoking effects. We have added this consideration to the revised Discussion section (see lines 325-330, page 9):

“Despite including smoking status as a covariate in the models for all participating cohorts in both studies,²⁹ we recognize that our ability to fully separate the effects of diet from smoking on CHIP risk may be limited due to our cross-sectional study design. As a result, some of the observed association between unhealthy diet and CHIP through epigenetic mechanisms may reflect residual smoking effects.”

29. Ma J, Rebholz CM, Braun KVE, et al. Whole Blood DNA Methylation Signatures of Diet Are Associated With Cardiovascular Disease Risk Factors and All-Cause Mortality. *Circ Genom Precis Med.* 2020;13(4):e002766. doi:10.1161/CIRCGEN.119.002766

References of Reviewer 1:

1. Stacey, S.N., et al., Genetics and epidemiology of mutational barcode-defined clonal hematopoiesis. *Nature Genetics*, 2023. 55(12): p. 2149-2159.
2. Phipson, B., J. Maksimovic, and A. Oshlack, missMethyl: an R package for analyzing data from Illumina's HumanMethylation450 platform. *Bioinformatics*, 2016. 32(2): p. 286-8.
3. Maksimovic, J., A. Oshlack, and B. Phipson, Gene set enrichment analysis for genome-wide DNA methylation data. *Genome Biol*, 2021. 22(1): p. 173.
4. Reed, Z.E., et al., Exploring pleiotropy in Mendelian randomisation analyses: What are genetic variants associated with 'cigarette smoking initiation' really capturing? *Genet Epidemiol*, 2024.
5. Dekkers, K.F., et al., Blood lipids influence DNA methylation in circulating cells. *Genome Biol*, 2016. 17(1): p. 138.
6. Bernabeu, E., et al., Refining epigenetic prediction of chronological and biological age. *Genome Medicine*, 2023. 15(1): p. 12.
7. Seale, K., et al., A comprehensive map of the ageing blood methylome. *bioRxiv*, 2023: p. 2023.12.20.572666.
8. Bernstein, N., et al., Analysis of somatic mutations in whole blood from 200,618 individuals

identifies pervasive positive selection and novel drivers of clonal hematopoiesis. *Nat Genet*, 2024. 56(6): p. 1147-1155.

Reviewer #2 (Remarks to the Author):

The authors have presented a thorough and well-written paper exploring the role of methylation in a variety of CHIP subtypes. They linked CHIP to alterations in specific CpGs through an epigenome-wide association study (EWAS) and then validated these changes using an in vitro model - a step that added considerable depth to their analysis. The authors then linked CHIP-associated CpGs with gene expression to explore the functional role of these methylation changes. Finally, they performed a Mendelian Randomisation Analysis to identify putative causal CpGs linked to cardiovascular outcomes. Overall, we found the methodology generally sound and our concerns are relatively minor. That said, we have a few comments which are outlined below:

1. The EWAS looks fine to us, but we're not experts in cross-ancestry analyses. An additional reviewer might be considered who is an expert in such fields.

Response: We thank the reviewer for this feedback. In this study, we used self-reported race as a demographic variable rather than genetically inferred ancestry (see line 139, page 5).

2. In the in vitro validation analysis, the authors focused only on the subset of CpG sites that were significantly associated with CHIP in both the EWAS analysis and nominally differentially methylated in the in vitro model of CHIP. 1) Why was this approach taken? In other words, why is it appropriate to ignore the CpGs that appear in only one set? 2) Similarly, while the directionality overlap in the subset of CpGs that change in both analyses looks generally quite good, why do we see such a small percentage of the EWAS hits captured in the experimental data in the first place?

Response: We thank the reviewer for these comments. We have added these points to the Discussion to clarify our approach and its limitations (see lines 297-307, page 9):

“In the *in vitro* validation analysis, we focused on the subset of CpG sites that were significantly associated with CHIP in the EWAS and nominally differentially methylated in the *in vitro* CHIP model to improve the validity of our findings and reduce the likelihood of false positives. By concentrating on overlapping CpG sites, we prioritized CpGs with a stronger potential biological relevance, as they were consistent across population-level and experimental settings. Importantly, despite the strong directional concordance in the subset of overlapping CpG sites, a small proportion of the CpGs identified from the meta-EWAS were captured in the *in vitro* model: ~14% (855/5990) for *DNMT3A*, ~6% (312/5633) for *TET2*, and ~2% (139/6078) for *ASXL1* CHIP. This may be because the *in vitro* system does not fully capture the complex *in vivo*

environment in which CHIP is influenced by various cell types, environmental factors, and systemic interactions, such as immune system interactions.”

3. *In the in vitro validation analysis, the \neg ASXLI-associated CpG sites showed no significant hits in ASXLI-engineered cells. We would like to see this validation failure discussed more clearly in the limitations of downstream analyses with this type of CHIP, i.e. does this affect the interpretation of any results?*

Response: In our revised manuscript, we have included further discussion on the lack of experimental validation for *ASXLI* CHIP EWAS findings and the resulting limited validity of our study’s downstream analyses with *ASXLI* CHIP (see lines 277-279, page 8):

“The lack of enrichment of *ASXLI*-associated CpGs in the *ASXLI*-engineered cells may limit the validity of the study’s downstream analyses with *ASXLI* CHIP.”

4. *In the “Association of DNA Methylation with Gene Expression and Pathway Analyses” section, it might be interesting to see how the directionality of gene expression changes aligns with methylation changes.*

Response: In our revised manuscript, we have included discussion of the concordance between directionality of gene expression changes with methylation changes in the Results section as follows (see lines 212-217, pages 6-7):

“The vast majority of the associations between methylation and gene expression changes are negative, i.e., decreased methylation changes are associated with increased gene expression changes or increased methylation changes are associated with decreased gene expression changes. For any CHIP, *DNMT3A*, *TET2*, and *ASXLI* CHIP, ~68% (317/467), ~71% (184/258), ~77% (224/293), and ~72% (168/234) of CpGs have a negative association between methylation and gene expression changes, respectively.”

5. *Additionally, also in the “Association of DNA Methylation with Gene Expression and Pathway Analyses” section, gene expression changes within 1Mb of the CpG in question were analyzed. This is quite a large distance – the authors may also want to consider smaller distances, e.g. 100kb, which may help increase the specificity of their results.*

Response: We appreciate this insightful suggestion. We have revised the manuscript by re-defining *cis* as within 100 kB rather than 1 MB. This adjustment was made with the goal of enhancing the specificity of the GO enrichment analysis of the eGenes identified in our eQTM analysis (see lines 206-222, pages 6-7). The text has been changed as follows:

“We analyzed the associations of CHIP-associated CpGs with changes in *cis* gene expression (expressed gene [eGene] within 100 kB of CpG) in 2115 FHS participants whose DNA methylation data and whole-blood RNA-seq data were available. At $P < 1 \times 10^{-7}$, we identified 467 significant *cis* CpG-transcript pairs for any CHIP, 258 for

DNMT3A CHIP, 293 for *TET2* CHIP, and 234 for *ASXL1* CHIP (Supplementary Tables 18-21 provide the full expression quantitative trait methylation (eQTM) results).¹¹ The vast majority of the associations between methylation and gene expression changes are negative, where decreased methylation changes are associated with increased gene expression changes or increased methylation changes are associated with decreased gene expression changes. For any CHIP, *DNMT3A*, *TET2*, and *ASXL1* CHIP, ~68% (317/467), ~71% (184/258), ~77% (224/293), and ~72% (168/234) of CpGs have a negative association between methylation and gene expression changes, respectively. For any CHIP, the top enriched GO terms related to lipid metabolism. eGenes associated with *DNMT3A* CHIP were enriched in cell motility and adhesion processes. For *TET2* CHIP, the top enriched terms related to immune processes, such as leukocyte differentiation. *ASXL1* CHIP eGenes were enriched in cellular and immune processes, including cell importation and antigen processing and presentation (Supplementary Tables 22-25).”

6. *The section “Association of DNA Methylation with Genetic Variants and Mendelian Randomization Analysis” needs some more detail (perhaps just in Methods) regarding the first two paragraphs. E.g. specifically, how were mQTLs linked with diseases?*

Response: In response to this reviewer comment, we have revised the manuscript to include additional details on the methods used to assess the association of DNA methylation with genetic variants, complex diseases, and traits. These details are now provided in the Methods section as follows (see lines 633-643, page 17):

“To annotate CHIP-associated CpGs and *cis*-mQTLs, we utilized both the EWAS Catalog (<https://www.ewascatalog.org/>)²³ and the GWAS Catalog (<https://www.ebi.ac.uk/gwas/>).²¹ The EWAS Catalog collected published CpG signatures for about 4000 traits and/or diseases. GWAS Catalog collected significant SNPs associated with thousands of traits and/or diseases. For the identified CHIP-associated CpGs, we matched these CpGs with reported trait-associated CpGs in the EWAS Catalog. To evaluate the enrichment of CHIP-associated CpGs for traits listed in the EWAS Catalog, we performed Fisher’s exact tests. We applied a Bonferroni-corrected significance threshold of $P = 1.24E-05$ ($0.05/4023$, accounting for 4023 traits in the EWAS Catalog). Additionally, to assess whether any *cis*-mQTLs of CHIP-associated CpGs demonstrated strong associations with human complex traits, we matched the *cis*-mQTLs against SNPs in the GWAS Catalog that were reported with $P < 5E-8$.”

7. *We would also like to see some more discussion of the potential limitations of the Mendelian Randomisation approach in identifying causal CpGs. In particular, do the authors believe that cis-mQTLs satisfy the “exclusion restriction” assumption of an instrumental variable? In other words, is it possible/likely that the cis-mQTLs might be affecting the outcome through pathways other than via the exposure? If so, how does this affect the interpretation of the results?*

Response: We thank the reviewer for raising this important point. In the revised manuscript we now elaborate in the Discussion section on the potential limitations of the Mendelian randomization analysis (see lines 385-390, page 11):

“Despite our attempt to minimize horizontal pleiotropy by excluding *cis*-mQTLs that also serve as *trans*-mQTLs and excluding CpGs with three or more independent instrumental variables through MR-Egger using a threshold of P value <0.05, the two-sample MR approach has known limitations.³⁷ Methods for detecting and addressing pleiotropy may be ineffective³⁸ and, thus, longitudinal and functional studies are needed to reinforce causal findings.”

In our MR analysis, the rationale for using *cis*-mQTLs as IVs is based on the concept that genetic variants near a CpG site (*cis*-mQTLs) influence DNA methylation at that site—a relationship well-supported by various biological studies. However, this approach has limitations, primarily because *cis*-mQTLs may not fully satisfy the "exclusion restriction" assumption, potentially leading to horizontal pleiotropy. To minimize such pleiotropy, we initially excluded *cis*-mQTLs that also serve as *trans*-mQTLs. For CpGs with three or more independent IVs (*cis*-mQTLs with LD < 0.01 within a 2 MB region around the CpG), we conducted MR-Egger to assess pleiotropy, where a non-zero intercept would suggest its presence; results with P values < 0.05 were excluded. Despite these measures, two-sample MR has recognized limitations, and current methods for detecting and addressing pleiotropy may be insufficient under certain plausible scenarios. For example, genetic instruments, especially for complex traits such as smoking, may exhibit horizontal pleiotropy, impacting multiple traits not causally related to the exposure and suggesting possible pleiotropic effects.¹ In our manuscript, we now acknowledge the importance of longitudinal and functional studies to strengthen causal interpretations.

38. Reed ZE, Wootton RE, Khouja JN, et al. Exploring pleiotropy in Mendelian randomisation analyses: What are genetic variants associated with 'cigarette smoking initiation' really capturing?. *Genet Epidemiol*. Published online August 4, 2024. doi:10.1002/gepi.22583

8. *The legend of Fig 3 doesn't seem quite right – it states “Dot plots of methylation status from -1.0 (no methylation) to 1.0 (complete methylation)”, but it seems it should probably refer to methylation change.*

Response: The legend of Figure 3 has been changed to “methylation change” in place of “methylation status” (see page 22).

Reviewer #3 (Remarks to the Author):

Reviewer #4 (Remarks to the Author):

RE: Kirmani et al.

Epigenome-wide DNA Methylation Association Study of CHIP Provides Insight into Perturbed Gene Regulation

Summary

Kirmani et al. present an epigenetic characterization of CHIP compiled from several large epidemiological sequencing cohorts. By looking at DNA methylation profiles from individuals with CHIP, the authors try to deduce some profiles that may link the mutations to regulation of epigenetic changes that drive some CHIP-related pathologies. There is also attempt to connect differentially methylated CpGs to downstream gene expression changes, although nothing is really significant.

The manuscript is essentially descriptive and several limitations impact the relevance of the study. It is not clear what the real conclusion from this analysis is or how it contributes to our understanding of how the mutant cells gain clonal dominance (which as outlined by the authors in the introduction is part of the motivation for the study). In the current state, it is uncertain what information can be obtained from this that advances the field of CHIP.

Major Points:

1. The genomic, DNA methylation and gene expression profiles are obtained from whole blood samples. The fraction of mutated cells in this sample will be highly variable and typically very low. While it is certainly impressive that certain DNA methylation patterns can be associated with DNMT3A, TET2, and ASXL1 mutations given this, how do the authors account for VAF of mutation with altered DNA methylation?

Response: In the revised manuscript, we now account for the possibility that VAF of CHIP driver gene mutation may influence DNA methylation by identifying associations between CHIP-associated CpG sites and VAF of mutation (see lines 391-398, page 11), as follows:

“To account for the possibility that the VAF of CHIP driver gene mutation may influence DNA methylation, we used a threshold of FDR <0.05 to detect associations between CpGs and VAF. There were no significant associations between CpG sites and VAF. Based on these results, we do not believe that VAF significantly influenced our DNA methylation findings. Notably, the sample size with available VAF information is limited. For example, in the FHS cohort, we have only 166 CHIP cases and cannot completely rule out the possibility that VAF may still have a modest impact on DNA methylation. Experimental studies or studies with larger sample sizes may be necessary to address the effect of VAF of mutation on DNA methylation.”

2. Given it is known that these common CHIP mutations alter the differentiation spectrum of mutant stem and progenitor cells, the cellular composition of each individual with different mutations is going to be highly variable. How can the authors account for hematopoietic cell

type specific DNA methylation differences? DNA methylation is essentially a measure of cell type identity. And the DNA methylation profile of a neutrophil is very different from a T-cell. Any changes in DNA methylation may simply reflect different proportions of mature hematopoietic cell types that went into the sequencing pool.

Response: We thank the reviewer for highlighting this important point. We recognize that common CHIP mutations can lead to variability in hematopoietic cell composition, which in turn influences DNA methylation profiles. To account for this, in all of our cohorts – Framingham Heart Study, Jackson Heart Study, Cardiovascular Health Study, and Atherosclerosis Risk in Communities, – cell type composition was included as a covariate in the statistical models (see Supplementary File 2). This adjustment allows us to better distinguish between methylation changes due to cell type proportions and those directly associated with CHIP mutations.

3. The authors engineer DNMT3A, TET2, and ASXL1 mutations into healthy CD34+ cells and then perform DNA methylation profiling. They mostly recover the DNA methylation profiles from the sequenced CHIP donors. It is unclear what additional value is gained from this experiment. How does this inform of the mechanism by which these mutations change DNA methylation profile and skew cell fate decisions?

Response: We have included further discussion on the value of profiling DNA methylation in engineered human CHIP cells beyond functional validation of the EWAS findings (see lines 289-296, page 8). The revised text is as follows:

“By incorporating an engineered reductionist system, we provide an orthogonal approach to confirm that the patterns observed in CHIP donors result directly from the somatic mutation. By engineering *DNMT3A*, *TET2*, and *ASXL1* mutations into healthy CD34+ cells and performing DNA methylation profiling, we can recapitulate the DNA methylation patterns seen in CHIP donors. Our study also establishes this reductionist system as a robust representation of the methylation phenotype. Future studies leveraging this system will enable more precise dissection of the causal relations between these mutations and changes in DNA methylation than would be possible from population-scale epidemiology data alone.”

4. There is a largely no correlation between altered DNA methylation patterns and changes in gene expression profile (line 211). Is there any information that can be obtained to inform how altered epigenomic profile of CHIP cells alters their function?

Response: The reported associations between CHIP-associated CpG sites and *cis* gene expression are all significant at $P < 1 \times 10^{-7}$. The GO enrichment analyses of the differentially expressed genes, however, were not significant after correction for multiple testing. In the revised manuscript, in response to comments from Reviewer #2, we re-defined *cis* eGenes as within 100 kB of the CpG, as opposed to 1 Mb, to potentially improve the specificity and significance of our eQTM gene ontology enrichment analysis results.

By considering a smaller distance between the CHIP-associated CpG and gene-level transcript, GO enrichment results for the differentially expressed genes are now significant at FDR < 0.05. These new results are included in the revised manuscript (see lines 205-222, pages 6-7), as follows:

“To understand how differentially methylated CpGs in association with CHIP might alter cellular function, we identified gene expression changes associated with CHIP-linked CpGs. We analyzed the associations of CHIP-associated CpGs with changes in *cis* gene expression (expressed gene [eGene] within 100 kB of CpG) in 2115 FHS participants whose DNA methylation data and whole-blood RNA-seq data were available. At $P < 1 \times 10^{-7}$, we identified 467 significant *cis* CpG-transcript pairs for any CHIP, 258 for *DNMT3A* CHIP, 293 for *TET2* CHIP, and 234 for *ASXL1* CHIP (Supplementary Tables 18-21 provide the full expression quantitative trait methylation (eQTM) results).²⁰ The vast majority of the associations between methylation and gene expression changes are negative, where decreased methylation changes are associated with increased gene expression changes or increased methylation changes are associated with decreased gene expression changes. For any CHIP, *DNMT3A*, *TET2*, and *ASXL1* CHIP, ~68% (317/467), ~71% (184/258), ~77% (224/293), and ~72% (168/234) of CpGs have a negative association between methylation and gene expression changes, respectively. For any CHIP, the top enriched GO terms related to lipid metabolism. eGenes associated with *DNMT3A* CHIP were enriched in cell motility and adhesion processes. For *TET2* CHIP, the top enriched terms related to immune processes, such as leukocyte differentiation. *ASXL1* CHIP eGenes were enriched in cellular and immune processes, including cell importation and antigen processing and presentation (Supplementary Tables 22-25).”

5. Line 77-79: “The novel genes and pathways linked to the epigenetic features of CHIP may serve as therapeutic targets for preventing or treating CHIP-mediated diseases.” This is very speculative. Please be more specific about what novel genes / pathways have been identified by this study for this purpose. And how might they serve as therapeutic targets in the future?

Response: In the revised manuscript we now provide more details about a specific novel gene, *FCRL3*, that may serve as a therapeutic target and how it may be targeted to treat *TET2* CHIP-mediated diseases in the Discussion (see lines 409-424, pages 11-12):

“For example, *Fc receptor-like protein 3 (FCRL3)* (cg17134153, $F_x = -5.5$, $P = 1E-113$) is the top differentially expressed gene for *TET2* CHIP. *FCRL3* encodes a type I transmembrane glycoprotein that is expressed by lymphocytes and plays a role in modulating immune responses.³⁹ Polymorphisms in this gene have been implicated in the pathogenesis of autoimmune diseases.^{39,40} A recent study demonstrated that *FCRL3* stimulation of regulatory T cells induced production of pro-inflammatory cytokines, including IL-17 and IL-26.³⁹ This finding suggests that *FCRL3* may play a critical role in mediating the transition of regulatory T cells to a pro-inflammatory phenotype and could potentially contribute to the increased inflammation observed among *TET2* CHIP

carriers.^{41,42} Additionally, Clark et al. identified *FCRL3* as a gene for which DNA methylation at the CpG site cg17134153 in CD4⁺ T cells likely mediates the genetic risk for rheumatoid arthritis.⁴³ Given that CHIP, including *TET2* CHIP, has been associated with rheumatoid arthritis (RA),⁴⁴ the regulation of *FCRL3* expression by methylation changes at cg17134153 may, in part, serve as the functional basis of the observed association between CHIP and RA. Further experimental studies are warranted to better understand how differential expression of *FCRL3* may impact *TET2* CHIP development and the pathogenesis of RA.”

Responses to Reviewers' Comments

The updated reviewer comments are in *italics*, and the new responses are in plain font.

Reviewer #1 (Remarks to the Author):

>>> Reviewer #1 Replies

Kirmani et al. have performed a meta-EWAS for CHIP in ~8k samples (450k and EPIC). CHIP was identified via Mutect2 in WGS in 3 cohorts (FHS, JHS, and CHS) and via Exome-seq in the ARIC cohort - with the requirement of a variant allele frequency (VAF) $\geq 2\%$ in peripheral blood and absence of hematologic malignancy. The EWAS was performed for the 3 major mutational subtypes (DNMT3A, TET2, ASXL1) and identified ~6k associated CpGs in each, with an overlapping total of $\sim < 10k$ at $p < 1E-7$. A sensitivity analysis increasing the VAF to 10% recapitulated ~78% of these CpGs.

The majority of the CpG effects were in trans - consistent with the global action of these mutated epigenomic machinery genes. The expected opposite directional effects of TET2 and DNMT3A mutants were observed, with predominately hypermethylation and hypomethylation, respectively.

The downstream effects of these CHIP-associated CpGs were interrogated, including identifying expression quantitative trait methylation (eQTM) and functionally via human hematopoietic stem cell (HSC) CRISPR models. High concordance between the meta-EWAS and human-edited CHIP HSCs was observed. Causal inference via two-sample Mendelian Randomization (MR) identified 261 CpGs associated with CVD traits and all-cause mortality.

In this work, the authors have extended previous analysis in a subset of these data (Uddin et al.) and include additional functional evaluation. I have no major methodological issues with the analysis but highlight some points for the authors to consider below.

Major

1. The authors state in the introduction that CHIP is associated with cardiovascular disease (CVD). They need to include comment/interpretation on the DeCODE findings that the association of smoking with CH itself solely drives this observation [1].

Response: We thank the reviewer for this suggestion. We have revised the manuscript to include discussion about the DeCODE study and their finding that smoking drove the association between CHIP and CVD.

The revised manuscript is changed as follows (see lines 348-370, page 10):

“Although smoking status was included as a covariate in the statistical models for all study cohorts, there could be residual confounding as smoking behavior may not be fully adjusted for in the analysis. Thus, smoking could still be driving the association between CHIP and CVD, as was reported in a recent study of CH.³² Of note, there are several

studies across diverse populations and in different settings that controlled for smoking as a covariate and also found an association of CHIP (a specific form of CH) with CVD. For instance, in a recent study, Diez-Diez et al.³³ clarified the directionality of the CHIP-CVD relationship with adjustment for smoking and concluded that CH confers an increased risk of developing atherosclerosis.

Additionally, the way that the DeCODE32 investigators ascertained “CH” may have contributed to their finding that CH was not associated with CVD, as it is distinct from the definition of “CHIP.” CH includes clonal events with known leukemic driver gene mutations, such as CHIP and mosaic chromosomal alterations (mCAs), and clonal events without clear driver genes. Given recent findings that each of these distinct classes of CH has unique phenotypic consequences,^{3,34} the lack of association between CH and CHIP reported by Stacey et al. may be due to the grouping of heterogeneous CH subtypes. Moreover, it has been previously demonstrated that small changes in the stringency with which CHIP is ascertained can have an outsized effect on downstream analyses. For example, Vlasschaert et al.³⁵ reported the importance of CHIP detection stringency in relation to CHIP-associated CVD risk. Specifically, more stringent criteria (≥ 5 supporting reads) were associated with CVD risk, while less stringent criteria (≥ 3 supporting reads) attenuated the association of CHIP with CVD. Their study provides an up-to-date and nuanced explanation of the CHIP-CVD relationship.”

>>> *Great answer to this – thanks.*

2. The Pathway Enrichment method employed does not appear to have corrected for the known DNAm array bias. A methodology that corrects for this needs to be implemented for accurate enrichment [2, 3].

Response: Thank you for the suggestion. In the revised manuscript, we have now corrected the GO enrichment analysis for known DNAm array bias, ⁵⁷ as recommended. Specifically, we used the missMethyl R package to adjust for this bias, and have updated the manuscript to replace the previous GO enrichment results with the revised analysis. Results have changed as follows: (see lines 193-204, page 6):

“For any CHIP, DNMT3A CHIP, and ASXL1 CHIP the enriched GO terms related to broad cellular developmental and organismal processes, while for TET2 CHIP the top GO terms related to cellular regulation and cell signaling (Supplementary Tables 14-17). For example, for any CHIP and the two most common CHIP subtypes, the top ten most significant ontologies included multicellular organism development, anatomical structure development, system development, and developmental process. For TET2 CHIP, the most significant ontology terms related to a cellular response to stimulus, regulation of cellular processes, and cell signaling. Notably, for the genes annotated to the 554 TET2-associated CpGs that were found to be demethylated, the top GO terms were enriched for cellular developmental and organismal processes such as multicellular organism development and system development – similar to the enriched GO terms for genes annotated to DNMT3A CHIP-associated CpGs (Supplementary Table 29).”

>>> *It is good that the authors have performed this non-biased reanalysis*

3. Could the authors comment further on the ~10% of TET mutant CpGs showing decreased methylation – was there any pathway and/or genomic loci enrichment for these findings or potential enrichment for false positives (e.g., how did the p-value distribution compare with the DNMT3A result)?

Response: Thank you for your insightful suggestion. We performed GO and pathway enrichment analysis on the 10% of TET2 mutant CpGs showing decreased methylation. The findings have been incorporated into the manuscript (see lines 200-204, page 6):

“Notably, for the genes annotated to the 554 TET2-associated CpGs that were found to be demethylated, the top GO terms were enriched for cellular developmental and organismal processes such as multicellular organism development and system development – similar to the enriched GO terms for genes annotated to DNMT3A CHIP-associated CpGs (Supplementary Table 29).”

To address the possibility of false-positive enrichment, we compared the p-value distributions of the TET2-associated CpGs with decreased methylation to those of DNMT3A-associated CpGs (see Response Letter Figure 1). This comparison revealed no discernible patterns in the p-value distributions, indicating distinct effects of TET2 CHIP and DNMT3A CHIP on overlapping CpGs. These findings suggest that the demethylated CpGs associated with TET2 mutations are unlikely to be enriched for false positives. Instead, the weak or inconsistent correlation observed between TET2- and DNMT3A-associated demethylated CpGs further supports the independent roles of these mutations in methylation regulation.

Figure 1. Comparison of P-value Distributions Between Demethylated TET2 CpGs and DNMT3A CpGs

>>> *Ok*

4. The ASXL1-associated CpG sites showed no significant hits in ASXL1-engineered cells. The authors speculated this may be due to the limited number of ASXL1 CHIP cases in the EWAS. Could the authors comment if there could be any other biological explanation for this?

Response: The revised manuscript includes comments on potential biological explanations for ASXL1-associated CpGs showing no significant enrichment in ASXL1-engineered cells in the Discussion section (see lines 277-288, page 8):

“The lack of enrichment of ASXL1-associated CpGs in the ASXL1-engineered cells may limit the validity of the study’s downstream analyses with ASXL1 CHIP. This observation may be due to the limited number of ASXL1 CHIP cases in the EWAS as well as several biological factors. ASXL1 mutations primarily affect histone modifications, particularly H2AK119 ubiquitination, which indirectly influences chromatin accessibility.²⁶ Recent

studies have shown that ASXL1 loss-of-function mutations increase chromatin accessibility, potentially resulting in individualistic methylation changes influenced by other genetic and environmental factors.²⁷ Furthermore, the effects of ASXL1 mutations on methylation might be temporally dynamic or cell-type specific, aspects not fully captured in our current experimental design. Future studies with larger sample sizes and longer observation periods may help elucidate the complex relationship between ASXL1 mutations and DNA methylation patterns in CHIP.”

>>> *Ok*

5. Across all CHIP cases/subtypes expressed genes associated with a CHIP-associated CpG were enriched for immune function. Whilst most of the cohort EWAS included cell-type proportion covariates - could uncorrected subtle effects in cell-type proportion due to clonal selection in these immune cell types be contributing to this enriched immune function?

Response: Thank you for raising this important point. Although we included cell-type proportions as covariates in this study, we cannot exclude the possibility that subtle uncorrected effects in cell-type composition due to clonal selection in immune cells could potentially contribute to this observed enrichment. In the revised manuscript, we acknowledge that while cell-type adjustments reduce these confounding effects, residual contributions from altered immune cell proportions remain possible. See the revised Discussion (lines 399-405, page 11):

“Last, although cell-type proportions were included as covariates for all cohorts, we cannot exclude the possibility that subtle uncorrected effects in cell-type proportions due to clonal selection in immune cells may contribute to the enrichment of immune function observed for TET2 and ASXL1 CHIP eGenes. While cell-type adjustments reduce confounding effects, residual contributions from altered immune cell proportions remain possible. Future studies investigating cell-type specific DNA methylation and gene expression may provide additional clarity on the impact of CHIP on immune gene expression”.

>>> *Ok great*

6. 261 CHIP-associated CpG were identified via 2-step MR to be putatively causally associated with CVD-related traits and/or all-cause mortality. Considering the DeCODE results - could the authors comment on the weakness of the 2-step MR approach, i.e., appropriate statistic thresholds, pleiotropy, etc - as even genetic MR analysis acknowledges the need for further rigour [4].

Response: We thank the reviewer for raising this important consideration. In our analysis, we used the inverse variance weighted (IVW) method for MR to explore potential causal relationships between CHIP-associated CpGs and CVD traits or all-cause mortality, using independent cis-mQTLs as instrumental variables. To minimize horizontal pleiotropy, we

initially excluded cis-mQTLs that also acted as trans-mQTLs. For CpGs with three or more independent IVs (defined as cis-mQTLs with LD < 0.01 within a 2MB cis-region of the CpG), we further assessed pleiotropy through MR-Egger, where a non-zero intercept would suggest potential pleiotropy, and excluded results with P values < 0.05. Despite these precautions, the two-sample MR approach has known limitations and current methods for detecting and addressing pleiotropy may be ineffective under some plausible scenarios. As noted in reference 4 from Reviewer #1, genetic instruments, especially for complex traits such as smoking, can show horizontal pleiotropy for multiple traits that are unlikely to be causally linked to the exposure, suggesting possible pleiotropic influences. We acknowledge that longitudinal and functional studies are critical for reinforcing causal findings and we now discuss these limitations in the revised manuscript (see lines 385-390, page 11):

“Despite our attempt to minimize horizontal pleiotropy by excluding cis-mQTLs that also serve as trans-mQTLs and excluding CpGs with three or more independent instrumental variables through MR-Egger using a threshold of P values <0.05, the two-sample MR approach has known limitations.³⁷ Methods for detecting and addressing pleiotropy may be ineffective³⁸ and, thus, longitudinal and functional studies are needed to reinforce causal findings.”

>>> *Ok great*

7. Regarding the FADS2 cg11250194 finding and the comment that hypomethylation of this CpG may be linked to higher triglyceride levels – was this CpG identified previously in any lipid-associated EWAS? [5]

Response: Yes, CpG site - cg11250194 – was identified in an epigenome-wide association study of lipid-related metabolic measures performed by Gomez-Alonso et al³⁰ (see lines 317-318, page 9):

“Moreover, cg11250194 was previously identified in an EWAS of lipid-related metabolic measures.³⁰”

>>> *Great good to include this info*

8. Although the EWAS included age as a covariate - a proportion of the common CpG changes identified in all 3 mutants could still be ageing and not CHIP mutant-related. What is the overlap with recent ageing meta-EWASs? [6, 7]

Response: There are 19 CpGs that are common across all three CHIP driver gene mutations – DNMT3A, TET2, and ASXL1 CHIP (Bonferroni-corrected $P < 1 \times 10^{-7}$) from our meta-EWAS of CHIP. When comparing these CpG sites with the CpGs from a recent EWAS of chronological age,¹ we found that none of the CpGs overlapped. We have included this new information in our revised Results section on lines 161-166, page 5:

“Although age was included as a covariate, there remains a possibility that common CpG

sites across all three CHIP driver genes may be related to age rather than CHIP mutation. To assess this, we compared the 19 CpGs that are common among all three CHIP driver gene mutations from our meta-EWAS of CHIP with the CpGs from a recent EWAS of chronological age in the Generation Scotland cohort¹⁷ (N=18,413). No CpGs overlapped, suggesting that common CpGs across all CHIP driver genes are related to CHIP mutation rather than age.”

>>> *Ok good to exclude this specifically*

9. As the authors acknowledge, issues around power affect the ability to evaluate less frequent CHIP-associated mutants. Could the authors include comment on potential ‘driverless’ CH caused by epigenetic changes alone as discussed by Bernstein et al. [8]

Response: We thank the reviewer for this comment. We have included in the revised Discussion section (lines 371-382, pages 10-11) comments on the potential ‘driverless’ CH caused by epigenetic changes as follows:

“Driverless CH is the occurrence of clonal expansions in blood without a known CHIP driver mutation and is estimated to drive the majority of clonal expansions in the elderly.³⁶ Bernstein et al.³⁶ identified regions within exome sequences that are under positive selection to identify additional driver mutations in whole blood (large clones >0.1) together with validation of positive selection in single cell-derived hematopoietic myeloid and lymphoid colonies. The inclusion of mutations in these fitness-inferred CH genes increases prevalence of CH by 18% in the UKBB cohort. In our study, CHIP was defined when an individual harbored at least one deleterious insertion/deletion or single nucleotide variant in any of the 74 genes that have been previously linked to myeloid malignancy at a variant allele frequency of at least 2%. Given the study's scope, we did not include driverless CH in our CHIP definition. Thus, CHIP prevalence in our study may be underestimated relative to studies that account for driverless clonal expansion.”

>>> *Ok, good to include*

10. In the Discussion the authors speculate that an unhealthy diet may be associated with CHIP through epigenetic mechanisms. Could this be confounded by concurrent poor diet with smoking?

Response: We thank the reviewer for raising this point. We included smoking status as a covariate in the models for all participating cohorts, which reduces the likelihood that smoking alone accounts for the observed association between CHIP and CpG sites. Ma et al.²⁹ also used models that adjusted for smoking status when identifying diet-associated CpG sites. However, we recognize that our cross-sectional study design may limit the ability to fully separate the effects of diet from smoking on CHIP risk, as cross-sectional analyses capture associations at a single time point without assessing how these exposures and their interactions may change over time. Consequently, some of the observed association between unhealthy diet and CHIP through

epigenetic mechanisms may still reflect residual smoking effects. We have added this consideration to the revised Discussion section (see lines 325-330, page 9):

“Despite including smoking status as a covariate in the models for all participating cohorts in both studies,²⁹ we recognize that our ability to fully separate the effects of diet from smoking on CHIP risk may be limited due to our cross-sectional study design. As a result, some of the observed association between unhealthy diet and CHIP through epigenetic mechanisms may reflect residual smoking effects.”

>>> *Ok great to qualify this*

Reviewer #2 (Remarks to the Author):

The authors have addressed our concerns.

Reviewer #3 (Remarks to the Author):

Reviewer #4 (Remarks to the Author):

The authors have done a thorough job at responding to previous reviewer comments. Technically they have answered all the concerns raised in the prior review.

However, there is a difference between statistical significance and biological significance. The power in this study comes from leveraging large patient groups to achieve high sample numbers required for robust statistical comparisons. However, the authors have still not clearly accounted for biological diversity. Although the statistical justification of cell type correction is robust and justified, how this plays out in a real world setting is not validated. Similarly, it is still dubious how evaluation of individual CpGs is meaningful given the complex nature of DNA methylation. Differentially methylated regions would be more meaningful. So while this is an excellent study leveraging a large patient populations, the real biological implications of this analysis remain unclear.

Response: We thank the reviewer for raising this point. In the revised Discussion section (see lines 400-409, page 11), we have included this limitation to our study as follows:

“While this study benefits from a large sample size, which allows for robust statistical comparisons, we acknowledge the distinction between statistical significance and biological significance. The statistical significance observed in this study does not necessarily equate to meaningful biological effects. The impact of cell type correction on downstream analyses, particularly in heterogeneous patient populations, has not been fully validated. Further research is needed to determine how this adjustment translates into biological outcomes. Additionally, our study evaluated individual CpG sites, however, differentially methylated regions consisting of several consecutive methylated CpGs have been shown to have important implications for disease pathogenesis.³⁷ Thus, studies exploring these broader methylation patterns are warranted to better capture the functional relevance of epigenetic signatures of CHIP.”